# Exploring non-retention in clinical trials: a meta-ethnographic synthesis of studies reporting participant reasons for drop out

Zoë C Skea, Rumana Newlands, Katie Gillies

## ABSTRACT

**Objectives** To undertake a meta-ethnographic synthesis of findings from primary studies reporting qualitative data that have explored participant-reported factors influencing non-retention within a clinical trial context.

**Design** A systematic search and meta-ethnography was conducted for published papers (from 1946 to July 2018) that contained qualitative data from trial non-retainers.

**Participants** We identified 11 studies reporting qualitative data from 13 trials. The studies were undertaken between 2008 and 2018. Each study included between 3 and 40 people who had dropped out from a trial, with findings from 168 people in total reported across the papers.

**Results** Emergent from our synthesis was the significance of trial non-retainers' perceptions around the personal 'fit' of key aspects of the trial with their personal beliefs, preferences, capabilities or life circumstances. These related to their own health state; preferences for receiving trial 'care'; individual capabilities; beliefs about or experiences of trial medication and considerations whether trial participation could be accommodated into their broader lives. All these factors raise important issues around the extent to which initial decisions to participate were fully informed.

**Conclusions** To improve retention in clinical trials, researchers should work to reduce the burden on trial participants both through the design of the intervention itself as well as through simplified data collection processes. Providing more detail on the nature of the trial interventions and what can be expected by 'participation' at the consenting stage may prove helpful in order to manage expectations.

## Strengths and limitations of this study

► Trial retention has recently been identified as one of the top three priorities for methodological research by UK trialists.

► Within the context of clinical trials, issues around retention have not received equal scrutiny compared with methodological questions about trial recruitment despite being arguably just as important for trial validity.

► Understanding the complex reasons why trial participants leave a trial after initially consenting is important if trialists are to be able to design effective intervention strategies to address the problem.

► To our knowledge, this is the first synthesis of key qualitative findings from studies exploring participants' perspectives of trial non-retention, which provides learning across their collective contributions.

► Our synthesis only included 11 eligible papers reporting findings across 13 trials, 5 of which were set within a mental health context and all of which were conducted in high-income countries. This could have issues for the transferability of findings.

Health Services Research Unit, University of Aberdeen, Aberdeen, UK

**Correspondence to**
Dr Katie Gillies;
k.gillies@abdn.ac.uk

## INTRODUCTION

Randomised controlled trials (RCTs) are integral for evidenced-based clinical decision making. Within the context of clinical trials, the focus of much methodological research in recent years has been on issues specifically relating to trial recruitment, including significant investigation into how to increase the numbers of prospective participants recruited.[1][2] A key focus of much of this research has been on trial participants' perspectives and experiences particularly around why they do or do not choose to consent to participate in clinical trials.[3–7] While issues relating to trial recruitment are undoubtedly important, issues around retention (ie, ensuring that trial participants remain in the trial to provide primary outcome data) have not received equal scrutiny in the literature despite being arguably just as important for trials in terms of ensuring that research questions are adequately answered.[2]

Trial retention was recently identified in the top three priorities for methodological research by UK trialists.[8] Most trials experience the issue of missing data often referred to as a 'loss to follow-up', 'attrition' or 'drop out' and this can bias the findings of a trial. Some recent quantitative surveys have identified participant characteristics (eg, age, gender, physical or mental health) or trial processes (eg, study duration or length and

relevance of outcome measures) as being potential predictors of trial retention.[9–11] However, these studies are small in size, often limited to a particular clinical context, and the items included in the surveys are often identified by researchers rather than asking participants what items should be included. In addition, they lack any in-depth exploration of the relevant issues affecting why participants withdraw, as reported by participants.

Understanding the complex reasons why trial participants leave a trial (either actively (eg, by requesting no further follow-up or purposefully not returning data) or passively (eg, forgetting to return a questionnaire or attend a clinic visit)) after initially consenting to participation is important, especially if those reasons are modifiable. This understanding of participant perspectives then becomes crucial if trialists are to be able to design effective intervention strategies to address the problem.

The approach of conducting in-depth qualitative research within the context of clinical trials is considered particularly useful for improving the evidence base for how trialists conduct them.[12] Indeed, this approach has been used widely to explore perspectives on trial recruitment both in terms of primary qualitative studies and secondary syntheses. To our knowledge, this is the first synthesis of key findings from studies exploring participants' perspectives of trial non-retention, which provides learning across their collective contributions. Our aim was to undertake a meta-ethnographic synthesis of findings from such studies and our specific research question was 'what influences non-retention in clinical trials'?

## METHODS
A systematic literature search and meta-ethnography was conducted (see online S1 ENTREQ Checklist). This meta-ethnography was undertaken in two parts. Our original systematic search and synthesis was undertaken in August 2016. To integrate potentially more recent relevant research, we undertook an update in July 2018.

Meta-ethnography essentially involves an 'interpretive and inductive' approach to synthesising studies.[13 14] Essentially meta-ethnography involves the process of 'translating' the findings of individual qualitative studies so that they can be considered in relation to one another with the aim of identifying and building new conceptual knowledge on a particular topic.[13 14] The process of 'translating' findings across studies can be either 'reciprocal' or 'refutational' depending on how individual studies relate to each other.[13]

### Searching and identification of relevant studies
A systematic search was conducted for published papers that contained qualitative data about trial participants' reasons for not completing some or all of the processes involved in a clinical trial after initially consenting to take part (which we describe as constituting non-retention). Search strategies were informed by previous studies[12] and are provided in online supplementary appendix 1. Seven electronic databases were searched by an information specialist: Embase, Ovid MEDLINE, PsycINFO, Cochrane Central Register of Controlled Trials, The Social Sciences Citation Index, Cumulative Index of Nursing & Allied Health Literature and Applied Social Sciences Index and Abstracts and covered papers published from 1946 to August 2016 (first search) and from August 2016 to July 2018 (updated search). Google Scholar and bibliographies of identified publications were also searched manually for additional potentially eligible papers.

For both searches, one author screened all titles and abstracts (RN for original search; ZCS for update) with a second author (KG) screening a random 10% sample. Eligible studies included those that used qualitative methods and contained qualitative data exploring any aspect of non-retention from the perspective of patient participants (recognising that non-retention might cover activities such as cessation of or withdrawal from the intervention(s), non-attendance at clinic visits, through to non-response to some or all follow-up questionnaires, etc).

### Analysis and synthesis
In order to collate and synthesise the available primary research, the seven steps of meta-ethnography as listed in online supplementary S1 box were followed. In summary, the three authors (ZCS, RN, KG) each read and systematically extracted data from the included papers, shared notes and discussed study findings and interpretations during a series of group meetings. The papers were initially organised in chronological order (but as inductive analysis progressed papers were grouped according to emerging themes) and we focused on the findings, concepts and themes used by the papers' authors generating a list of key categories. We used a standard form which summarised the main themes, information regarding methods and any other important information relating to the context of the research within each study (some of these data are illustrated in online supplementary S1 table). Although we initially organised papers chronologically in this table, we used it to facilitate a series of further group discussions around emerging issues (see online supplementary S2 table). As inductive analysis progressed, we grouped and discussed our data according to the five key emerging themes (see online supplementary S3 table). In line with the process of undertaking a meta-ethnography, primary data or 'first-order constructs' (quotations from study participants who had not completed any or some of the various trial processes) and authors' interpretations of these data ('second-order constructs') were extracted, compared and contrasted between studies (enabling us to produce a 'reciprocal translation'), and organised into themes to facilitate the development of new insights or a 'line of argument'.[13]

### Study quality
One author (ZCS) undertook a quality assessment of each of the papers included in the synthesis. This was based on

the Critical Appraisal Skills Programme (CASP) criteria,[15] which was used to appraise the identified primary studies and consider their inclusion into the synthesis (see online supplementary S1 CASP checklist). Questions developed by the CASP have been used previously for appraising the quality of studies for inclusion in meta-ethnography.[16–20]

## Patient and public involvement

This research was done without patient involvement. Patients were not invited to comment on the study design and were not consulted to develop patient relevant outcomes or interpret the results. Patients were not invited to contribute to the writing or editing of this document for readability or accuracy.

## RESULTS
### Description of studies

The database search produced 1431 abstracts for the initial search and 697 abstracts for the update (see online supplementary S1 Figure and S2 Figure for details). We only included studies that provided data about reasons for non-retention from the included study participants and/or in the authors' reflections. In all, 11 papers met our inclusion criteria (8 were identified from the initial search and 3 from the update). The focus and key study characteristics for the 11 included papers are outlined in online supplementary S1 table. The identified papers were conducted in seven countries (the UK, the USA, Australia, Sweden, The Netherlands, Denmark and Spain) and discussed non-retention in 13 separate trials. Six of the papers focused solely on reasons for non-retention,[21–26] with the remaining five also considering reasons for consenting,[27] non-consenting[28 29] and retention.[30 31] The findings in this synthesis relate to the data from non-retainers only. Each study included between 3 and 40 people who had dropped out from a trial, with findings from 168 people in total reported across the papers. As can be seen from online supplementary S1 table, the setting of the trials in which the qualitative research was embedded included a range of clinical contexts such as: mental health problems[21 24 26]; mental health problems and cancer[23]; problem drinking[22 25]; type 1 diabetes[30]; diabetes, chronic obstructive pulmonary disease (COPD), heart failure or social care needs[28]; severe ankle sprains[31]; asymptomatic atherosclerosis[29]; neurodevelopment disorders[27] and osteopenia.[27] As expected, the clinical context differed as did the interventions under investigation and included: telehealth equipment or telecare devices[28]; web-based psycho-educational/cognitive therapy-based support tools[21 22 24 26]; antidepressant medication and/or cognitive behavioural therapy[23]; exercise[25 30]; various mechanical ankle supports[31]; aspirin[29]; melatonin[27] and bisphosphonate risedronate or vitamin D analogue 1-alpha-hydroxycholecalciferol.[27]

Findings were presented from trial non-retainers both before outcome data had been collected (eg, those who withdrew from the intervention) and/or during the follow-up when outcome data were being collected—in other words, papers included a mix in terms of non-retention behaviour (see online supplementary S1 table for a summary of non-retention behaviour, that is, non-adherence to intervention, non-return of questionnaires). For example, eight studies reported aspects related to non-adherence to trial intervention: three of these reported cessation of trial medication[23 27 29] (for both trials); five reported cessation of treatment therapy sessions[21–24 26]; one reported cessation of use of telehealth equipment or telehealth devices[28] and another reported non-completion of study workbooks.[21] Two studies reported non-return of follow-up questionnaires.[22 31] For two studies, non-retention behaviour was unspecified.[25 30] Three of the 11 studies appeared to have included only the views of those who had dropped out of the active intervention arms of the trial.[23 24 28] For four studies, it was unclear whether data were from intervention or control groups[26 27 29 31] and only four studies specifically stated that they included views of both those in the intervention and control groups.[21 22 25 30]

Nine of the 11 studies used semi-structured interviews to collect data from people who had withdrawn from the main trial[21 23–28 30 31]; one used a combination of focus groups and interviews[29] and another distributed a questionnaire that contained various open-ended response options[23] (NB: only the qualitative data are reported and referred to in this paper). Although some papers provided gender, age and/or demographic details for participants taking part in the trial in question, as can be seen from online supplementary S1 table, this information was less comprehensive for those who had dropped out of the trial. Where participant characteristic information was provided in the original studies, we have included this at the end of the quotes presented to illustrate findings.

### Key themes from the synthesis

Our grouping of first-order and second-order constructs across the 8 initially identified papers resulted in 14 subthemes. During the process of translating themes from each of the individual studies (ie, comparing and contrasting across studies), these subthemes were then grouped and categorised into five broad key themes which characterised the main considerations and features that appeared to influence non-retention in the trials under investigation (see online supplementary S2 table). For the three subsequently identified papers, we repeated the various stages of meta-ethnography—in essence comparing for 'fit' and checking for any additional themes.[32 33] For the update, we attempted to follow the 'extend and renovate the house' approach,[33] which involves examining the newly included studies to establish whether they add new concepts or contribute to existing ones. During this process, we were confident that concepts identified in the later three papers supported and complemented our originally identified five key themes (from the original eight studies) with no new concepts emerging.

These themes were: (1) perceptions of current health state in relation to specific aspects of the trial; (2) the 'fit' of

aspects of the trial with individual preferences for care and support; (3) the compatibility of aspects of trial processes with individual capabilities; (4) concerns about or experiences of trial medication and (5) considerations around the extent to which trial participation could be appropriately accommodated into individuals' broader lives.

As these theme labels suggest, within them they accommodate a spectrum of views or experiences.

The five broad key themes identified as influencing participants' non-retention in clinical trials are illustrated with example data in online supplementary S2 table. In online supplementary S2 table, primary study participant quotes illustrating first-order constructs are displayed in italics, and primary study author interpretations illustrating second-order constructs are presented in bold text. In the rest of this paper, primary study participant quotes are displayed in italics.

### Influences on participant non-retention in clinical trials: a line of argument

Expressed below is our 'line of argument', which is organised into themes to facilitate the development of cumulative insights (online supplementary S3 figure conceptually illustrates the line of argument developed from the synthesis). These themes appear to be weighed up during the participant's involvement in the trial and set alongside the complex inter-relationship between self and trial processes/procedures and ultimately impact on their retention in the trial. Overall, our argument emphasises the significance of trial participants' perceptions around the 'fit' of key aspects of the trial (intervention and trial processes) with their personal beliefs, preferences, capabilities or life circumstances. These factors (which were not necessarily mutually exclusive) related to beliefs about their own health state, preferences for how they wanted to receive care, their individual capabilities, beliefs about or experiences of trial medication and also considerations around the extent to which trial participation could be appropriately accommodated (or not) into their broader lives. All of these were set against the overall backdrop of the balance between their sense of self and the trial processes and procedures—this providing the overarching explanation for the influence on retention in trials. Implicit within several of these identified factors is the suggestion that there may have been deficits within the initial trial consenting process which led to participants (who subsequently withdrew) not being fully informed or at least not realising what the trial expected of them and what they could expect of the trial. These findings are discussed in more detail below and arranged across five key themes.

### Perceptions of current health state in relation to specific aspects of the trial

This theme describes how aspects of the trial might not be right for people as individuals. For example, across 8 of the 11 studies a key influence on decisions to discontinue trial participation appeared to relate to perceptions of either being 'too well' to warrant further engagement with the trial[21–23 25 28 29 31] or struggling with the compatibility of aspects of the trial, particularly the interventions or ways outcomes were assessed, with their personal sense of self.[21 22 25 28–30] Conversely, other participants described periods of feeling too unwell to be able to engage appropriately in trial processes.

#### Being too well to engage further with trial processes

Some participants cited a belief that they had suitably recovered part way through a particular trial as a reason for discontinuing trial medication and/or problem solving treatment exercises.[21–23 25] For some, this was also linked to not wanting to be reminded about health issues that they considered to be over:

> *I just don't want to be reminded of the alcohol thing, because I actually think it's over*[25] (female, 30–68 years, alcohol use disorder)

> *Things really improved for me…I just felt really good and didn't really feel like I had that much to offer in regard to finding out more about it*[21] (female, 30–39 years, bipolar disorder, control group)

> *I have been sufficiently helped*[22] (no gender/age details, problem drinker, intervention group)

Participants also cited recovery as a reason for not completing and returning all the required follow-up outcome assessment questionnaires,[28] (severe ankle sprains) perhaps highlighting here the importance at the consenting stage of making sure participants are fully informed about the value of sustained engagement throughout the duration of the trial (even if they feel they are no longer personally benefiting from that engagement).

#### Lack of compatibility with personal sense of self

Sometimes reasoning around trial withdrawal related to participants' struggle to accommodate aspects of the trial with their personal sense of self at the time,[25 28 29] suggesting that the intervention challenged their sense of self somehow. Again perhaps indicating the importance for initial trial recruitment consultations to include adequate discussions about the nature of the study intervention and also what will be expected of participants in terms of engagement with them. For example, a belief that they could self-manage or cope well enough without the need to engage with the trial support intervention[28] (self-care intervention to facilitate support for self-management in ageing populations); a belief that they were too overweight and unfit to participate in a group exercise intervention[25] (exercise intervention for people with alcohol use disorder); a belief that they had adequately managed their condition thus far without the need for any medication[29] (aspirin for asymptomatic atherosclerosis) and also non-acceptance of a diagnosis among those newly diagnosed[21 30] (with bipolar disorder; with type 1 diabetes) as a reason for not relating to (or seeing any value in) the study interventions:

(Discussing the need to keep active rather than monitoring his health indoors using telehealth equipment) *You've got (to have) the will power…if you can't do it I am finished. If I wouldn't have that I'd be, I'd be stuck inside here you know, and looking through the window like…I throw myself in the garden and everything. Everything I do I'm working on, I cook myself dinners and everything.*[28] (Male, 85 years, COPD)

*I think if it had been medication that I needed to take, I would have taken it*[29] (male, 72 years, stopped taking aspirin medication)

*If you're taking a lot, it knocks the hell out of your stomach…Given the choice, I'd rather not take medication full stop*[29] (male, 55 years, stopped taking trial medication for asymptomatic atherosclerosis)

*I wasn't ready to accept the illness. At that stage after diagnosis I wasn't willing to change my life according to the programme.*[21] (Male, 18–29 years, bipolar disorder, control group)

*Don't think it kind of really sank in as to what I'd been diagnosed with … It had kind of hit me and I wasn't really dealing with having it …* [30] (female, 19–55 years, type 1 diabetes)

If trial participants believed that the trial did not fit with their personal sense of self, this was also linked to an emotional response. For example, feelings of guilt and shame that they were too overweight and unfit to participate in a group exercise intervention[25] (exercise intervention for people with alcohol use disorder).

### Being 'too ill' to be able to engage appropriately with trial interventions

Conversely, within all of the papers focussing on interventions for mental health conditions, and in one paper focussing on people newly diagnosed with type 1 diabetes, participants described being 'too ill' to be able to engage appropriately in trial processes.[21 23 24 26 30] Reasons discussed in this context related to feeling either too fragile, depressed, too manic or too emotional/stressed at certain times to be able to complete the required intervention tasks (eg, e-health intervention and associated workbook activities; cognitive behavioural therapy; taking blood samples) and also a concern that engagement with the intervention could act as a 'trigger' in terms of exacerbating anxiety symptoms:

*I was feeling that the therapy wasn't going to help me with my problems. I thought it could lead me to be even more anxious and that it wasn't going to be beneficial for me. So, I felt that I was going to waste my time if I continued*[26] (no gender details, 21–59 years, people with a range of serious mental health problems)

*I did not cope with the exercises. I did them at the start but it gradually became more difficult to complete them.…particularly the breathing exercises. I got a bit dizzy and it increased my feelings of anxiety*[24] (no gender or age details, generalised anxiety disorder)

*The biggest problem I have with my bipolar disorder is consistency; when I'm down I can't even brush my teeth or get up in the morning. So doing an education programme with workbooks was beyond me*[21] (female, 18–29 years, bipolar disorder, Bipolar Education Program group)

*I often go walking when having highs because I have to keep moving, so I didn't want to sit at a computer*[21] (male, 40–49 years, bipolar disorder, bipolar education program (BEP) + informed supporter (IS) group)

As with the earlier subthemes in this section, emotional influences were also woven through this perception of being 'too ill' to engage with the trial. One study pointed to the 'emotional impact of the cancer diagnosis' as being an influential factor linked to participant drop out.[23]

### The 'fit' of aspects of the trial with individual preferences for care and support

Across 8 of the 11 studies another important influence in decisions to discontinue trial participation appeared to relate to the fit of aspects of the intervention with preferences for how participants wanted to receive care and support,[21–26 28 31] implicitly suggesting that the initial trial consenting process may have been suboptimal in key ways. Participants in these trials discussed how aspects of the design of the interventions were not individualised or tailored enough to be helpful and others commented on interventions being either too technical, too physically demanding, too intensive or conversely too basic:

*I needed a therapy that could better address what I felt. It didn't give me a specific answer to my worries.*[26] (No gender details, 21–59 years, people with a range of serious mental health problems)

*I would have liked to have more of a personal contact, it became a little distant everything, to do on the internet, because it is so heavy stuff, it's nice to meet a real person when you're working with heavy things like this*[24] (no gender or age details, generalised anxiety disorder)

*I wanted something more about me specifically, as opposed to talking about general issues*[21] (male, 40–49 years, bipolar disorder, BEP group)

*The information in the modules was too general and too limited*[21] (male, 18–29 years, bipolar disorder, BEP group)

Some other participants simply indicated that they had been unhappy or dissatisfied or 'not comfortable' with the treatment they had received, although specific reasons were not provided within the included studies.[22 23 31]

### The compatibility of aspects of trial processes with individual capabilities

Across 3 of the 11 studies,[23 24 31] the extent to which aspects of the interventions were deemed to be appropriately 'pitched' at the individual emerged as being of importance. For example, participants cited attention

problems and limited reading and writing skills as a reason for withdrawing from internet delivered cognitive behavioural therapy[24] or as a reason for non-response to follow-up questionnaires,[31] with participants in one of these studies stating that they felt unintelligent because of their inability to understand.[24] Communication and cultural issues were also cited as reasons for the discontinuation of problem solving treatments,[23] suggesting that these issues would benefit from greater consideration and discussion at the consenting stage:

> *I thought that it was too much to read, and I cannot read anything at all that I need to remember or learn. It goes in here and out there (pointing at the ears)*[24] (no gender or age details, generalised anxiety disorder)

### Concerns about or experiences of the trial medication

Across two of the eight studies which were set within trials testing drug interventions[27 29] , concerns about the study medication were cited as reasons for discontinuing with trial participation. These included concerns that the trial drug(s) were not properly tested/licensed,[27] concerns that the trial medication could negatively interact with other prescribed medication,[29] through to citing a dislike of taking too much medication[29] or that the trial medication tasted offensive.[27] Constructs within this key theme again suggest potential issues with the informed consent process and highlight the importance of discussions about the purpose of any trial, the nature of trial medications and also the implication for participation of having certain comorbidities, linking back into the complex inter-relationship between self and trial processes/procedures:

> *It just scared me when it said not to be given to children under 20…I didn't understand they weren't licensed for children…and that's what I thought it was, just to see if it worked, not to actually like so then it could be licensed*[27] (mother of child in trial for young people aged 4–18 years with rheumatic diseases)

> *again I found that I had stomach problems with the tablet so I assumed that it must be the aspirin…*[29] (female, 63 years, stopped taking trial medication for asymptomatic atherosclerosis)

> *…and they discovered I had heart fibrillation…After that I'd to go on warfarin you see, so that's why I had to drop out because warfarin and aspirin just don't agree*[29] (female, 77 years, stopped taking trial medication for asymptomatic atherosclerosis)

> *I didn't think I really wanted to go on at the start but mum and dad persuaded me to. And so…when I was getting really fed up I just said 'No I don't want to' because I didn't like the taste (of the medicine)*[27] (POP trial; young person 11–14 years)

### Considerations around the extent to which trial participation could be appropriately accommodated into their broader lives

Aside from issues relating to beliefs about current health state, individual capabilities, preferences for care and concerns about side effects, participants also discussed how decisions to discontinue with trial participation related to other life 'events' that tended to take priority over or made it hard for them to engage fully with the various demands of the trial.[21–26 30 31] These factors appeared less directly related to the nature of the trial interventions themselves and more about the challenges of life in general (with one study[31] suggesting that these people could be classed more as 'happy' rather than 'unhappy' non-responders, in the sense that non-retention may be related to aspects out with the trial itself). Reasoning here involved trading off trial participation with competing priorities and ranged from events such as work or family, moving to another country, exams, pregnancies, postal strikes, etc and more generally simply daily routines that got in the way. Within one study,[25] the importance of existing social networks was highlighted, with some participants citing a lack of support from family members as a reason for discontinuing trial participation. Within this theme participants also sometimes cited 'laziness' or 'forgetfulness' as reasons for why they had either not completed trial interventions or had not responded to follow-up questionnaires with some apparently being unaware that they were being considered as 'drop outs' by study researchers:

> (Discussing cessation of therapy sessions/non-completion of study workbooks) *I didn't have the time, and with everything else, it wasn't a priority*[21] (female, 18–29 years, bipolar disorder, control group)

> (describing why they did not return a follow-up questionnaire) *Do you know what…laziness I'm just gonna put it down to that*

> Researcher: *OK and em it wasn't because you were disgruntled about part of the project?*

> *Definitely not no*[31] (no gender/age details given, severe ankle sprains)

> (discussing cessation of problem solving treatment sessions) *Did I drop out? No, I didn't dropout. I became busy and I figured I started missing calls.*[23] (Female, no age details, cancer and depression)

## DISCUSSION
### Principal findings

Our meta-ethnographic synthesis sought to explore factors that influence non-retention within clinical trial contexts. We identified 11 studies (reporting qualitative data from 13 trials) that explored participant-reported reasons for not completing any or some of the various trial processes (after initially consenting to take part). What emerged from our analysis was the importance of trial participants' perceptions about the personal compatibility of key aspects of the trial with their personal beliefs, preferences, capabilities or life circumstances. These factors related to their own health state, preferences for how they wanted to receive care, their

individual capabilities, beliefs about or experiences of trial medication and also considerations around the extent to which trial participation could be appropriately accommodated or not into their broader lives (conceptually illustrated in online supplementary S3 figure). Our synthesis has also highlighted that people's reasoning around dropping out of a trial can be described as being more or less 'active' in nature, with some people in our synthesis not even realising that they were being considered by the researchers as trial 'drop outs'.[23] All these factors raise important issues around the extent to which initial decisions to participate were fully informed and illustrate the importance for trial recruiters of ensuring that prospective participants are made aware of what the trial will entail and also what will be expected of them in terms of full participation.

Quantitative surveys have tended to investigate non-retention in the context of non-response to follow-up questionnaires. These studies have identified either participant characteristics or trial processes as being potential predictors of trial retention.[9–11] While these studies have a place, it is arguably difficult to influence some of these previously identified factors influencing retention as they may not be modifiable, for example, age or study duration. Our synthesis of more in-depth qualitative data has usefully built on these findings and has enabled a more nuanced understanding of key issues of relevance (which are potentially modifiable) relating to non-adherence to interventions and non-return of follow-up questionnaires. Participant characteristics as well as trial processes are of importance but we have also demonstrated that there can be a complex inter-relationship between the two. For example, a perception that the nature of the intervention negatively affects one's mental health can be of importance as can perceptions about the nature of the intervention in relation to perceptions of self or in relation to personal preferences for care and support. Furthermore, the compatibility or otherwise of various trial processes with individual capabilities can have implications for retention. Reasons given for not completing various trial processes were not necessarily mutually exclusive, but were rather a synergistic combination of factors that could apparently work towards trial non-retention. Our findings also highlight that some participants' behaviour around leaving a trial could be described as being more or less 'active' in nature (eg, stopping trial medication because of a concern around side effects (active) versus simply not remembering or being too busy to return a questionnaire (passive)). This is an important finding and one that has not been given due consideration in previous literature to date. People's views and life situations can change over time, all having the potential to impact on their retention within a trial. Furthermore, different types of trials are likely to present particular challenges in terms of their potential for non-retention. It could also be that certain types of reasoning might be more or less modifiable and easier to address particularly if they can be anticipated upfront during the trial design stage.

A recent study exploring reasons why people declined trial participation at the consent to recruitment stage has found that most declined at the outset because they judged themselves ineligible or not in need of the specific trial therapy in question.[34] The study authors suggest that to improve recruitment to trials the most successful interventions are likely to be the ones that focus on patients' assessments of their own eligibility and their potential to benefit from the trial treatment, rather than reducing trial burden per se. In our synthesis, we found that perceptions around eligibility and assessments regarding potential to benefit from the trial treatment were also considerations for people who had initially decided to join but who had subsequently ceased to engage. For example, this included those who felt that they had recovered such they did not need to engage further[21–23 25] and those who felt they could manage sufficiently well without engaging with the intervention.[21 28 29] However, in the context of non-retention, it is worth considering issues around trial burden (eg, interventions that might be perceived to be too technical or too demanding given a person's health state) as well as issues around preference for particular styles of care and support and acknowledging that the specific intervention and, or, the ways outcomes are assessed has to be compatible within the context of trial participants' broader lives. In other words, issues around reducing trial burden is of importance, both in terms of the intervention itself and also the ways that follow-up data are collected.

We know from previous syntheses of qualitative studies focusing on trial recruitment that people often choose to enter into trials in the hope of gaining some help for themselves from the intervention (even if they also state they are doing so for altruistic reasons, ie, to benefit research more generally), so-called 'conditional altruism'.[7] Some participants in our synthesis described perceptions around feeling too ill to continue taking part or feeling suitably better such that trial engagement was no longer warranted.[21–25] This perception of improvement in health would appear to resonate with the concept of conditional altruism in the sense that people might cease participation if they perceive their condition improves or conversely deteriorates, such that in effect their benefit for self has been realised and their continued participation is no longer warranted. Our finding here is perhaps exaggerated in trials with a mental health context (which applied to 6 of the 13 included trials), where diagnoses can adversely affect people's ability and inclination to initially take part in research.[34–36] We have shown that this issue also has relevance for retention in such trials as people's health states can be particularly vulnerable to fluctuation.[34] A recent meta-synthesis of factors affecting recruitment to depression trials[37] indicated that decisions can depend on issues relating to: perceptions of health at the time of invite; attitudes towards the research and trial interventions and the demands of the trial. Our synthesis has shown that some of this reasoning might also have the potential to impact on non-retention in those who are successfully recruited. Furthermore, previous research

has suggested that the therapeutic alliance can have an impact on adherence to treatment.[38] Within the papers included in our synthesis, this was not something that was discussed per se. However, as one of our key themes illustrate, some decisions to discontinue trial participation appeared to relate to the fit of aspects of the intervention with preferences for how participants wanted to receive care and support. Within this, some trial non-retainers stated that they had wanted more face-to-face personal contact with, for example, a therapist. This comparable finding could suggest that the underlying beliefs, preferences and expectations about trial participation are not explored and unpacked fully during trial consenting discussions.

## Strengths and limitations

We recognise that different review teams may interpret qualitative data in slightly different ways due to pre-existing world views or expertise across research areas. However, a strength of undertaking a meta-ethnographic synthesis of findings from studies providing qualitative data on factors influencing non-retention within clinical trials is that it has allowed us to gain important new shared insights into factors that seem to affect retention across a range of trial contexts—to our knowledge, this is the first study to have synthesised these primary studies in this way. Through synthesising, we have been able to pull insights from across studies, providing learning from their collective contributions. However, our systematic search identified only 11 eligible papers reporting findings across 13 trials, 5 of which had a mental health context and all of which were conducted in high-income countries. This in part perhaps reflects the difficulties researchers face in gaining access to the views of those who disengage with research. Furthermore, unlike surgical trials, all the included papers incorporated within their trials, interventions that participants could choose to discontinue engaging with (eg, taking drugs; stopping CBT, etc). While qualitative research does not usually intend to be generalisable, it is nevertheless important to consider the transferability of our findings to other clinical trial contexts and settings and one could argue that participants within mental health trials, surgical trials or trials that involve surrogate/proxy consent including those involving children[27] might face very different issues and challenges regarding retention. Although we were reassured that the key themes we identified had resonance across the included papers to a greater or lesser extent and so are likely to be important considerations within a range of clinical trial contexts, some influences on trial non-retention are likely to be more trial specific than others (eg, concerns about trial medication).

We carried out a quality assessment of the 11 included papers (see online supplementary S1 CASP Checklist). Although all papers had study aims that were amenable to investigation via qualitative means and all included qualitative data, some were deemed richer than others in terms of data and insights (ie, first-order and second-order

constructs). Arguably, this made undertaking a meta-ethnography in this context quite challenging as the number of studies and volume and/or quality of available data can affect depth of analysis. For example, one paper only reported qualitative data from open-ended questionnaire response options,[22] and two were deemed less useful in terms of presenting only very limited qualitative data (both first-order and second-order constructs).[22,23] Nevertheless, we did feel that they provided some helpful insights that usefully built on the findings of the other papers. Furthermore, despite some variation in the overall level of quality, due to the small number of included studies we felt it was more important to retain any relevant findings rather than disregard based on study quality. In doing so, we would argue that all 11 papers contributed useful elements to the collective whole and enabled us to develop our line of argument in terms of the issues of importance regarding trial non-retention.

## Practice implications

The way in which a trial is presented to individuals needs to take account of the influencing factors we have identified in this synthesis. While not all the factors we identified are modifiable, there influence needs to be recognised. We would argue that trialists need to think carefully about how the design of their trial might contribute to non-retention and that there is potential to modify trial design to improve retention.

To improve retention in clinical trials, researchers should work to reduce the burden on trial participants both through the design of the intervention itself as well as through simplified data collection processes. Providing more detail on the nature of the trial interventions and what can be expected by 'participation' (ie, when and how data will be collected) at the consenting stage may prove helpful in order to manage expectations.

Some people in our synthesis appeared to be unaware that they were being considered as trial non-retainers by the study researchers. This raises the question of participants' understanding of the importance of remaining in a trial for its duration (ie, completing the intervention and the outcome assessments) and its implications for the study in question. This finding is supported by a recent study of patient information documentation from UK NIHR funded trials that has highlighted that withdrawal and retention are poorly described and that statements about the value of retention are infrequent.[39] If trialists want to improve retention to clinical trials then there is an argument for giving the importance of completing the trial more prominence in patient information materials (and also during any trial recruitment discussions).

Our synthesis also potentially highlights the issue of people's awareness or lack thereof of what the trial interventions would entail. If trialists want to improve retention then this suggests an argument for also providing more detail on the nature of the trial interventions at the consenting stage in order to manage expectations. We know from previous literature that patient/public

involvement at the front end of trial design tends to be extremely limited if indeed it happens at all.[40 41] Given some of the key factors we found as being influential for non-retention, one could speculate that some early and meaningful patient/public involvement would be particularly useful (eg, for ensuring that aspects of the trial are user-friendly and as compatible as possible with the target population's likely preferences and capabilities).

### Implications for research

A Cochrane review investigating interventions to improve retention in trials has highlighted that most strategies to improve retention have focused on trying to improve follow-up questionnaire response.[42] Of these interventions, only monetary incentives have been shown to have a significant effect on return of questionnaires and the review highlighted that very few studies included trial participants in their design or development.[42] Our synthesis has demonstrated that there may be a range of issues relevant to trial participants that influence non-retention, which may not be amenable to modification by 'incentives' or other interventions that fail to consider participants during development.

As mentioned previously, qualitative methods to improve recruitment to trials is now recognised as a well-established methodology built into the design and delivery of large publicly funded clinical trials. The Qunitet Recruitment Intervention (QRI) is gathering momentum across a range of trials and Clinical Trials Units as a mechanism to unpack many of the nuances around how participants are recruited to RCTs.[43] Many of the approaches in the QRI are directly transferable to questions about retention. For example, how it is discussed in consultations and trial paperwork, what do stakeholders (trial participants and trial staff) report as the barriers and facilitators to retention, and work in this area could prove fruitful for minimising non-retention in ongoing RCTs. However, despite there being a clear need for more research in the context of trial retention, we also recognise the inherent challenges for researchers in obtaining the necessary ethical approvals for this type of research (particularly as current recruitment materials for trial participants tend to emphasise prospective participants' right to withdraw without given any reasons, etc). Therefore, development of shareable resources to facilitate regulatory approvals may be an important contribution for the trials methodology community.

Finally, given that synthesis was based on a sparse data set, with 5 of the 11 included studies focused on qualitative research within mental health trials, there is certainly scope for more good quality, rigorous primary studies exploring the barriers and enablers to trial retention from a participant's perspective across a range of clinical specialties and trial design types. Interestingly, our search did not identify any studies that had explored reasons for trial participants' non-attendance at trial follow-up visits. Ideally, future studies should consider and explore all aspects of trial process relevant for retention, including completion and return of data (and its mode of delivery or collection), and attendance at follow-up visits. A recent prioritisation exercise for research into trial retention has now identified the top 10 unanswered questions for trial retention.[44] Many of these top 10 questions lend themselves well to enquiry by qualitative research methods and priorities should be focused here.

## CONCLUSIONS

Our systematic literature search and synthesis has highlighted that there is very little published qualitative literature exploring participant-reported reasons for non-retention in clinical trials. Researchers have already called for 'a science of recruitment' in recognition that recruiting for science (eg, trials) is not currently underpinned by an evidence base around the factors which might have the potential to impact on recruitment.[1] This is undoubtedly important but we would also argue that we need to develop a parallel focus on 'a science of retention' if we are to start to be able to tackle the very real issue of non-retention in clinical trials. Our qualitative synthesis (of a small set of studies) feeds into this relatively undeveloped science and has shed some important light on the factors that might influence non-retention in clinical trials—factors that have implications both for practice and for further research. Taken together, the findings presented here and the subsequent implications for practice and research highlight the critical need to plan for retention as much as for recruitment during trial design and not treat it like the overlooked trial conduct 'Cinderella'.

**Acknowledgements** The authors would like to thank Cynthia Fraser for her assistance in helping develop the database searching strategies.

**Contributors** KG conceived the study idea. RN screened all titles and abstracts in initial search; ZCS screened all titles and abstracts in updated search with KG screening a random 10% sample from both searches. ZCS, RN and KG conducted the data analysis and ZCS wrote the initial and subsequent manuscript drafts. All the authors contributed critically to discussions about interpretation of data and revisions of manuscript drafts. All the authors approved the final version.

**Funding** ZCS was supported by a core grant from the CSO (reference CZU/3/3) and a Wellcome Trust Institutional Strategic Support Fund award (reference RG12724-18). RN was supported by a Wellcome Trust Institutional Strategic Support Fund award (reference RG12724-18). KG was supported by an MRC Methodology Research Fellowship (MR/L01193X/1).

**Competing interests** None declared.

**Patient consent for publication** Not required.

**Provenance and peer review** Not commissioned; externally peer reviewed.

**Data sharing statement** This is a review of published studies which are available to access through the relevant journals.

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
