## [Reviewer comments · BMJ Open]

ARTICLE DETAILS

TITLE (PROVISIONAL)	Exploring non-retention in clinical trials: A meta-ethnographic synthesis of studies reporting participant reasons for drop out
AUTHORS	Skea, Zoë; Newlands, Rumana; Gillies, Katie

VERSION 1 – REVIEW

REVIEWER	Athene Lane University of Bristol, UK
REVIEW RETURNED	29-Mar-2018

GENERAL COMMENTS	This is an interesting and novel analysis which warrants publication. One of the major limitations is that there were few studies identified and that most of these were in mental health trials, which may be real or artefactual related to the search terms ability to pick out pieces of research. Were any efforts made to use papers reporting relevant research to see their citations as a way of identifying additional research? I was unclear why the single arm trial was an exclusion. I was unsure that evidence was presented that effective PPI would improve retention, so this might be removed or reworded as more a discussion point? The abstract would benefit from the search dates. There seemed to be no reference to the impact of the therapeutic alliance on retention yet other research has suggested that it can have an impact on outcomes and adherence so would seem to overlap here?
---

REVIEWER	Jutta Bleidorn Hannover Medical School,, Institute for General practice
REVIEW RETURNED	27-Apr-2018

GENERAL COMMENTS	Please add information why the literature review ended in August 2016, which means that potential interesting new publications are not included. Implications and practice implications should focus more on relevant topics, to allow the reader to identify relevant themes quickly.
--

REVIEWER	Tim Rapley Northumbria University, UK
REVIEW RETURNED	18-Jun-2018

GENERAL COMMENTS	BMJ Open – Meta ethnography retention
---------------------------------------

	Overall, this is a nice paper exploring a good idea. P5 - I'd be tempted to report quality appraisal earlier in results section, rather than in discussion P5: What do you actually mean by 'identifiable data'? P6: You begin to raise this in the discussion – around inter-relationship - but I'm interested in making it clear in the results section about to what extent these are 'single factors' that influence retention, or rather how these are (reported as or not) multiple factors that influence a specific person. So, is it that, one is enough per se for some, or rather, one issue means others come into focus as they become related for some, or even, some are tipping point ones, that all other issues are workable/liveable, but when that is added it is too much etc. P6: I'm also interested in how this might not be about being 'fully informed' ((and this raises a question about how someone can ever be 'fully' informed about such issues – how do we make sense of how to operationalize an 'adequate discussion' given variety of contexts here)), or not even about 'not realizing', but rather they were aware of at the time, but not something they focused on as they did not feel relevant to them at the time. So we could see element of retention as, in part, an issue of temporality, change over time here – which you hint at later on discussion – some of these factors are very emergent and contingent. Results: I really don't feel the way you present your data actually works at the moment. Generally we have some text, then a list of quotes (and sometimes some more text to close section). I'm not sure what you gain from having multiple quotes, especially in list form, for most sections at the end of them. Do we really need two that make the same point (let's take section 1A, what does that really add here – why offer two, when makes same point?). So, give the readers a better flow, only use long form of quote if essential, tie it to the specific point your are making (so, show example at the point in argument) otherwise embed short sentence/phrase in text as part of flow. In 1B, the third quote at first read, makes less sense given your argument prior, but when embedded in narrative make some sense (but could be read as tied to emergent side-effects factor, not sense of self) and – albeit it again, we have potential issues here – was he not informed per se, did not realize, or felt at time of recruitment that would try this out anyway? What is interesting here is the phrase, 'given the choice' and how this relates to recruitment discussions. Results: I'm interested in seeing what this data set says about the process of saying 'no more'. Do you get any real sense about how this actually came about – as in, people saying, 'they did not tell me about that at initial recruitment conversation' etc – do people report on this as mis-understanding at that time point, lack of information, lack of focus from there part at that time etc. Do you get a sense that this, for some, difficult work – be they difficult decisions, difficult conversations etc – how reticent are people about saying no more – clearly some do this by defaults by not returning questionnaire or not coming to appointment. How are
--	--

others involved in this process – I know from trials I've been involved in, that at times, a specific health professional may play a part, so they may visit a GP to get a second opinion on whether a trial medication is tied to an emerging problem – or a family member may share concerns and so outline the problems they see etc

P13: I need to be more convinced about not including the later papers – and note, your reference for 'no guidance' is not tied to the right paper. I really do not feel you need guidance – the original 1988 text was quite sparse in some places - and I'm not sure it needs to be that specific. How would you treat this as additional data in any study? The key question is, does your synthesis still hold given the data? I'm not talking about 1/2nd order, but rather the synthesis? Do any new issues emerge from your five points – do you need more than those five - or is it just, maybe, a few modifications inside? I'm not saying you need to, but I want to know whether your conceptual model holds.

Abstract/P14: We could 'speculate' around role of (well designed and enacted) PPI and impact on retention – but not sure this needs to be in the abstract – as at the moment, it assumes a lot about the process.

And this ties to two issues I note above. Throughout the article we have a narrative of fully informed consent and it appears that the data you are working with tells us very little about the process of how this actually happens – you are assuming that this is tied to the recruitment conversation/documents. Clearly, these studies report on reasons given, but this needs to be embedded in (accounts about) why those reason happen. So, from reading your paper, one argument for further research is exploring why this happens (or actually, rather, how this h

BMJ Open – Meta ethnography retention

Overall, this is a nice paper exploring a good idea.

P5 - I'd be tempted to report quality appraisal earlier in results section, rather than in discussion

P5: What do you actually mean by 'identifiable data'?

P6: You begin to raise this in the discussion – around inter-relationship - but I'm interested in making it clear in the results section about to what extent these are 'single factors' that influence retention, or rather how these are (reported as or not) multiple factors that influence a specific person. So, is it that, one is enough per se for some, or rather, one issue means others come into focus as they become related for some, or even, some are tipping point ones, that all other issues are workable/liveable, but when that is added it is too much etc.

P6: I'm also interested in how this might not be about being 'fully informed' ((and this raises a question about how someone can ever be 'fully' informed about such issues – how do we make sense of how to operationalize an 'adequate discussion' given variety of contexts here)), or not even about 'not realizing', but

	rather they were aware of at the time, but not something they focused on as they did not feel relevant to them at the time. So we could see element of retention as, in part, an issue of temporality, change over time here – which you hint at later on discussion – some of these factors are very emergent and contingent. Results: I really don't feel the way you present your data actually works at the moment. Generally we have some text, then a list of quotes (and sometimes some more text to close section). I'm not sure what you gain from having multiple quotes, especially in list form, for most sections at the end of them. Do we really need two that make the same point (let's take section 1A, what does that really add here – why offer two, when makes same point?). So, give the readers a better flow, only use long form of quote if essential, tie it to the specific point you are making (so, show example at the point in argument) otherwise embed short sentence/phrase in text as part of flow. In 1B, the third quote at first read, makes less sense given your argument prior, but when embedded in narrative make some sense (but could be read as tied to emergent side-effects factor, not sense of self) and – albeit it again, we have potential issues here – was he not informed per se, did not realize, or felt at time of recruitment that would try this out anyway? What is interesting here is the phrase, 'given the choice' and how this relates to recruitment discussions. Results: I'm interested in seeing what this data set says about the process of saying 'no more'. Do you get any real sense about how this actually came about – as in, people saying, 'they did not tell me about that at initial recruitment conversation' etc – do people report on this as mis-understanding at that time point, lack of information, lack of focus from their part at that time etc. Do you get a sense that this, for some, difficult work – be they difficult decisions, difficult conversations etc – how reticent are people about saying no more – clearly some do this by default by not returning questionnaire or not coming to appointment. How are others involved in this process – I know from trials I've been involved in, that at times, a specific health professional may play a part, so they may visit a GP to get a second opinion on whether a trial medication is tied to an emerging problem – or a family member may share concerns and so outline the problems they see etc P13: I need to be more convinced about not including the later papers – and note, your reference for 'no guidance' is not tied to the right paper. I really do not feel you need guidance – the original 1988 text was quite sparse in some places - and I'm not sure it needs to be that specific. How would you treat this as additional data in any study? The key question is, does your synthesis still hold given the data? I'm not talking about 1/2nd order, but rather the synthesis? Do any new issues emerge from your five points – do you need more than those five - or is it just, maybe, a few modifications inside? I'm not saying you need to, but I want to know whether your conceptual model holds.
--	--

	Abstract/P14: We could 'speculate' around role of (well designed and enacted) PPI and impact on retention – but not sure this needs to be in the abstract – as at the moment, it assumes a lot about the process. And this ties to two issues I note above. Throughout the article we have a narrative of fully informed consent and it appears that the data you are working with tells us very little about the process of how this actually happens – you are assuming that this is tied to the recruitment conversation/documents. Clearly, these studies report on reasons given, but this needs to be embedded in (accounts about) why those reason happen. So, from reading your paper, one argument for further research is exploring why this happens (or actually, rather, how this happens) – and once that is done, we may have a much better sense of how best to intervene.
--	---

REVIEWER	Emma France University of Stirling, Scotland, UK
REVIEW RETURNED	26-Jun-2018

GENERAL COMMENTS	The manuscript covers an important topic for evidence-based health care - trial retention. As a result of a qualitative systematic review, the authors have provided some useful findings and conclusions for improving trial retention. There are however some important weaknesses in the manuscript which should be addressed. **Introduction & methods** 1. Why was the chosen qualitative synthesis methodology selected as an appropriate methodology compared to other qualitative synthesis methodologies? Meta-ethnography is suited to the synthesis of conceptual data – i.e. studies which have developed explanatory concepts, rather than those which just describe what people said/did. It looks like the 8 included studies did not achieve a conceptual level of analysis but provided fairly thin descriptive data making it very difficult to conduct a meta-ethnography and to achieve 'third order' or higher level interpretations. **Methods** 2. P 5. The authors say: 'The papers were initially organised in chronological order (but as inductive analysis progressed papers were grouped according to emerging themes).' If each paper contributed to a number of themes how could the whole paper be grouped according to a single theme? How were papers grouped by emerging themes? 3. P 5. The authors say 'we focused on the findings, concepts and themes used by the papers' authors generating a list of key categories.' This is confusing/ambiguous what does the part 'generating a list of key categories' mean? *Study Quality* 4. P 5. What is a 'brief' quality assessment? How did you do this and how was CASP used for this e.g. only specific questions used?
--

	Why did you decide to do a quality appraisal given they are controversial? The authors have not given details of the results of the quality appraisal. 5. There are some ambiguities surrounding the methods in S1 box and it deviates from the meta-ethnography methodology. On the basis of the information provided, what you have done resembles a thematic synthesis or thematic analysis rather than a meta-ethnography. 6. S1- the issues around QA being contentious/ having a lack of consensus should be referenced. 7. In S1 Box, the headings used are similar to but different from Noblit and Hare's 7 phases of meta-ethnography. It would be better to be consistent and use all of Noblit and Hare's labels unless there is an important reason for not doing so, Phase 1 = getting started, Phase 2 = deciding what is relevant to the initial interest, Phase 4 should be 'determining how the studies are related' Phase 5 is 'Translating the studies into one another' is and Phase 6 is 'Synthesizing translations'. These phases do overlap and can run in parallel but there are some distinct characteristics. 8. Reading the studies - Did all authors read FULL papers or just sections of papers? 9. Phase 4 has not been done in line with the meta-ethnography methodology – it is about more than just 'identifying' themes – it involves exploring the relationships between the studies, how they are similar and different. This can be done at the level of concepts/themes from the studies, the underpinning theories, the topic focus, the participants' characteristics, and so on. 10. P28. 'Translating the findings of each study into those of the others' Authors have not told us what the 14 sub-themes were or how these became 5 broad key themes. Grouping themes – how were they grouped, on what basis? What aspects of the themes from each of the individual studies did you compare and contrast? 11. P28. 'Synthesising the findings ' The description here does not resemble phase 6 of the meta-ethnography methodology (synthesising translations). In this phase you are looking to see if there are common types of translations from phase 5 and to see if some translations or concepts from studies can encompass others. Under phase 6 you could also explain how the 'line of argument' was arrived at. **Results ** 12. It is not clear what is a novel finding and what's taken directly from the papers so it is not clear if the authors have come up with anything new compared to the findings of the synthesised papers. There needs to be more transparency in what was in the papers and how they each contributed to the findings. Which papers supported and contributed to the 'themes'? For example, you could give a table showing the themes, findings or concepts that were found in each paper.
--	--

	13. P 6. The authors have not described - What were the 14 subthemes? How (using which criteria/ process) were they then grouped into 5 key themes? This synthesis appears to involve the re-grouping and re-categorising of themes/concepts from the synthesised papers rather than a new interpretation: 'Our initial grouping of first and second-order constructs across the 8 papers resulted in 14 sub-themes. [...]these were then grouped and categorised into 5 broad key themes' (PAGE 28). The themes resulting from the synthesis are descriptive and appear largely based on first order constructs (participants' views and experiences) rather than on second order constructs (the interpretations of the authors of the primary studies). 14. I cannot see any evidence that phase 6 of the meta-ethnography methodology has been conducted or a convincing line of argument synthesis produced (e.g. page 28). It resembles an aggregative review rather than an interpretive review. Meta-ethnography is an interpretive synthesis methodology. There are a number of published guides to selecting a qualitative synthesis methodology e.g. Gough, Thomas, & Oliver. (2012); Noyes & Lewin In Noyes et al (editors). Cochrane handbook, 2011; Paterson (2011) in Hannes & Lockwood; Booth et al 2018 RETREAT guidance. 15. S2 table is referred to but is missing. 16. Description of Studies 'Findings were presented from trial non-retainers both before outcome data had been collected (e.g. those who withdrew from the intervention) and/or during the follow up when outcome data was being collected' This is unclear and would benefit from re-wording- findings from where? Do you mean that the papers contained both of these kinds of data, either/or? 17. The studies were very diverse trials (ankle support/ psycho-education program/aspirin etc) - were retention reasons different? 18. Figure - how was this figure arrived at? It does not seem to reflect the themes that were developed e.g. sub-optimal informed consent is not a theme but is prominent in the figure. It probably should be a main theme as it's an important finding for trial retention and for ethical trial conduct. 19. The authors write: 'Our initial grouping of first and second-order constructs across the 8 papers resulted in 14 subthemes. During the process of translating themes from each of the individual studies into those of the others (i.e. comparing and
--	--

	contrasting across studies) these sub-themes were then grouped and categorised into 5 broad key themes which characterised the main considerations and features that appeared to influence non-retention in the trials under investigation.' 19a. How (using which process/criteria) were 1st & 2nd order constructs grouped? 19b. What were the 14 subthemes? 19c. How (using which criteria/ process) were they then grouped into 5 key themes? 20. Were there any refutational/contradictory findings, especially considering the heterogeneity of the included studies e.g. possible cultural differences and different focuses of the trials? 21. page 7. What the authors refer to as line of argument (LOA) appears to be 5 separate descriptive themes not an overarching theory or explanation. For example, how the LOA explains non-retention is under-developed. Some of the themes (experiences of side effects, care preferences, capabilities) are neither about pre-existing beliefs nor life circumstances so are not adequately covered in the LOA. (QA) using CASP. Even if you don't exclude studies on the basis of methodological concerns, if you do a QA, you should use this in some way e.g. to inform your discussion, using it to explain variation in findings e.g. describe which papers added most/more/little to the themes. **Discussion** 22. Page 12. The authors need to consider the review and synthesis process in the strengths and limitations and also the limitations of the study accounts/papers synthesised- e.g. missing data on participant characteristics, lack of conceptual findings. International studies were included - is this a strength or a limitation? 23. Page 13 – the authors say 'we were reassured that the key themes we identified had resonance across all of the 8 included papers to a greater or lesser extent and so are likely to be important considerations within a range of clinical trial contexts.' However, this is not reflected in the results– for the theme 'The compatibility of aspects of trial processes with individual capabilities' – only 3/8 studies contributed and for the theme 'Concerns about or experiences of the trial medication' – only 2/8 studies were relevant and so contributed. 24. Discussion, page 13 - the authors refer to quality appraisal but don't tell us in the results which papers were rich and how they scored on CASP- this could be added as an additional online file. The authors say 'some were deemed richer than others in terms of data and insights (i.e. first and second order constructs).' How was richness judged? Which papers were rich? Did the richer papers contribute more to the themes? 25. Page 13- It looks like the rate of publication of relevant articles is increasing rapidly since 4 new relevant publications were identified as being published between Aug 2016 and October 2017:
--	--

	'Our original systematic literature search was undertaken in August 2016. Since undertaking and expressing the synthesis within this paper we have updated our search (to October 2017) and have identified 4 additional potentially relevant papers [33,34,35,36].' This indicates that there are likely to be even more relevant articles now 8 months on that could usefully add to the synthesis which originally identified only 8 articles and which might fill gaps in the data/analysis. 26. There is recent guidance on why & how to update a qualitative synthesis/ meta-ethnography. The author's statement is incorrect: 'there is currently no guidance on how to do this for meta-ethnographies.' (page 13) Please see: France, E. F., Wells, M., Lang, H., & Williams, B. (2016). Why, when and how to update a meta-ethnography qualitative synthesis. Systematic reviews, 5(1), 44. The findings of this synthesis are potentially useful. If the authors can revise their description of their methodology to more accurately reflect the analysis they present then I think this paper can make a useful contribution to the trial methods literature. The recent rapid increase in relevant publications would merit an update to include the more recent papers in the synthesis.
--	--

VERSION 1 – AUTHOR RESPONSE

Reviewer(s)' Comments to Author:

Reviewer: 1

Reviewer Name: Athene Lane

Institution and Country: University of Bristol, UK

Please state any competing interests or state 'None declared': None declared

Please leave your comments for the authors below

This is an interesting and novel analysis which warrants publication. One of the major limitations is that there were few studies identified and that most of these were in mental health trials, which may be real or artefactual related to the search terms ability to pick out pieces of research. Were any efforts made to use papers reporting relevant research to see their citations as a way of identifying additional research?

Response: Thank you for your comments. We are glad that you found this an interesting and novel analysis. We can confirm that our database search was comprehensive and that all identified publications reference lists were searched manually for potentially eligible papers. In addition, we have updated our original search and as a result identified a further 3 papers.

I was unclear why the single arm trial was an exclusion.

Response: The study exploring drop out from a single arm trial was excluded as inclusion criteria for the study specified a minimum of two arms (i.e. an intervention and a comparator or another

intervention) and allocation to group through randomisation. These inclusion criteria were set as it was assumed (and supported by evidence in the literature) that drop out is more problematic in randomised studies comparing more than one intervention.

I was unsure that evidence was presented that effective PPI would improve retention, so this might be removed or reworded as more a discussion point?

Response: We have removed this sentence from the abstract. However, the point is mentioned in the discussion section.

The abstract would benefit from the search dates.

Response: Search dates have now been added to the abstract.

There seemed to be no reference to the impact of the therapeutic alliance on retention yet other research has suggested that it can have an impact on outcomes and adherence so would seem to overlap here?

Response: This is an interesting point. Indeed, previous research has suggested that the therapeutic alliance can have an impact on adherence to treatment. Within the papers included in our synthesis, this was not something that was discussed per se. However, as one of our key themes illustrate, decisions to discontinue trial participation appeared to relate to the fit of aspects of the intervention with preferences for how participants wanted to receive care and support. Within this, some trial non-retainers stated that they had wanted more face-to-face personal contact with for example, a therapist. We have now added detail about this and a reference to the discussion section.

Reviewer: 2

Reviewer Name: Jutta Bleidorn

Institution and Country: Dep. of General Practice, Medical School Hannover, Carl-Neuberg-Str.1, D-30625 Hannover

Please state any competing interests or state 'None declared': non declared

Please leave your comments for the authors below

Please add information why the literature review ended in August 2016, which means that potential interesting new publications are not included.

Response: We have now updated our search to end of July 2018.

Implications and practice implications should focus more on relevant topics, to allow the reader to identify relevant themes quickly.

Response: We have now added a short section to the start of the 'practice implications' section to highlight the themes identified in the synthesis.

Reviewer: 3

Reviewer Name: Tim Rapley

Institution and Country: Northumbria University, UK

Please state any competing interests or state 'None declared': None declared

Please leave your comments for the authors below
BMJ Open – Meta ethnography retention

Overall, this is a nice paper exploring a good idea.

P5 - I'd be tempted to report quality appraisal earlier in results section, rather than in discussion

Response: Thank you. We have now added the completed CASP Checklist as a supplementary file and have mentioned this in the methods section. We have also added more detail regarding our assessments of quality of the included studies to the discussion section.

P5: What do you actually mean by 'identifiable data'?

Response: We accept that the word identifiable data may cause confusion and have removed and edited to read "We only included studies that provided data about reasons for non-retention from the included study participants and/or in the authors' reflections."

P6: You begin to raise this in the discussion – around inter-relationship - but I'm interested in making it clear in the results section about to what extent these are 'single factors' that influence retention, or rather how these are (reported as or not) multiple factors that influence a specific person. So, is it that, one is enough per se for some, or rather, one issue means others come into focus as they become related for some, or even, some are tipping point ones, that all other issues are workable/liveable, but when that is added it is too much etc.

Response: Indeed, we agree that it's possible (and probably likely) that there could be just one or multiple reasons why someone drops out of a trial – hence our statement in the discussion section to this effect. However, our observations here were implicit as we only had access to the available published data from the 11 studies (and this was hard to disentangle from the included data).

P6: I'm also interested in how this might not be about being 'fully informed' ((and this raises a question about how someone can ever be 'fully' informed about such issues – how do we make sense of how to operationalize an 'adequate discussion' given variety of contexts here)), or not even about 'not realizing', but rather they were aware of at the time, but not something they focused on as they did not feel relevant to them at the time. So we could see element of retention as, in part, an issue of temporality, change over time here – which you hint at later on discussion – some of these factors are very emergent and contingent.

Response: This is a good point. Yes, we agree that some of these factors are likely very emergent and contingent. People's views/situations change over time and this might have an impact on their retention within clinical trials. As you say, we hint this in the discussion as our observations here were implicit, but we have now added an additional sentence re. this within the discussion.

Results: I really don't feel the way you present your data actually works at the moment. Generally we have some text, then a list of quotes (and sometimes some more text to close section). I'm not sure what you gain from having multiple quotes, especially in list form, for most sections at the end of them. Do we really need two that make the same point (let's take section 1A, what does that really add here – why offer two, when makes same point?). So, give the readers a better flow, only use long form of quote if essential, tie it to the specific point your are making (so, show example at the point in argument) otherwise embed short sentence/phrase in text as part of flow.

Response: We chose to construct our paper like this given it is a synthesis of primary studies. As we were dealing with multiple studies we wanted to illustrate the spread of data relating to our 5 key

themes across the 11 included papers to show areas of convergence and highlight areas of divergence across the studies.

In 1B, the third quote at first read, makes less sense given your argument prior, but when embedded in narrative make some sense (but could be read as tied to emergent side-effects factor, not sense of self) and – albeit it again, we have potential issues here – was he not informed per se, did not realize, or felt at time of recruitment that would try this out anyway? What is interesting here is the phrase, ‘given the choice’ and how this relates to recruitment discussions.

Response: We agree that our data raises a potentially important issue around informed consent. We made observations in the manuscript about people perhaps not being as informed as they could be at the consenting stage. Our interpretations of withdrawers’ quotes (presented in the main manuscript and S2 Table) illustrate (or at least strongly hint) that participants discovered aspects of the intervention post trial entry which made them inclined to withdraw. Again, these interpretations were based on our observations of the data as we only had access to the available published data from the 11 studies. These studies explored reasons for drop out as opposed to focussing on the process of recruitment and how this may or may not have impacted on retention.

Results: I’m interested in seeing what this data set says about the process of saying ‘no more’. Do you get any real sense about how this actually came about – as in, people saying, ‘they did not tell me about that at initial recruitment conversation’ etc – do people report on this as mis-understanding at that time point, lack of information, lack of focus from there part at that time etc. Do you get a sense that this, for some, difficult work – be they difficult decisions, difficult conversations etc – how reticent are people about saying no more – clearly some do this by defaults by not returning questionnaire or not coming to appointment. How are others involved in this process – I know from trials I’ve been involved in, that at times, a specific health professional may play a part, so they may visit a GP to get a second opinion on whether a trial medication is tied to an emerging problem – or a family member may share concerns and so outline the problems they see etc

Response: You make a very interesting point. Within our data, some people had apparently dropped out of the trial very early on (e.g. being newly diagnosed and having to deal with too much around the time of trial entry), whereas for others drop out had apparently happened at a later stage, as indicated by non-response to a follow up questionnaire for example. The data provided in the papers on this was incomplete and so it made it difficult for us to make more conclusive statement on this, but we have now integrated a sentence into the discussion to reflect that issues around retention will not necessarily be static – people’s views and life situations can change over time.

P13: I need to be more convinced about not including the later papers – and note, your reference for ‘no guidance’ is not tied to the right paper. I really do not feel you need guidance – the original 1988 text was quite sparse in some places - and I’m not sure it needs to be that specific. How would you treat this as additional data in any study? The key question is, does your synthesis still hold given the data? I’m not talking about 1/2nd order, but rather the synthesis? Do any new issues emerge from your five points – do you need more than those five - or is it just, maybe, a few modifications inside? I’m not saying you need to, but I want to know whether your conceptual model holds.

Response: We have updated our search to end of July 2018 and have identified and integrated findings from an additional 3 papers. For the 3 subsequently identified papers, we repeated the various stages of meta-ethnography - in essence comparing for ‘fit’ and checking for any additional themes (Lang 2013, France 2016). During this process, we were confident that concepts identified in the later 3 papers supported and complemented our originally identified 5 key themes (from the original 8 studies) with no new concepts emerging – and that our conceptual model still holds.

Abstract/P14: We could 'speculate' around role of (well designed and enacted) PPI and impact on retention – but not sure this needs to be in the abstract – as at the moment, it assumes a lot about the process.

Response: We have removed this sentence from the abstract. However, the point is mentioned in the discussion section.

And this ties to two issues I note above. Throughout the article we have a narrative of fully informed consent and it appears that the data you are working with tells us very little about the process of how this actually happens – you are assuming that this is tied to the recruitment conversation/documents. Clearly, these studies report on reasons given, but this needs to be embedded in (accounts about) why those reason happen. So, from reading your paper, one argument for further research is exploring why this happens (or actually, rather, how this happens) – and once that is done, we may have a much better sense of how best to intervene.

Response: This is a very good point. Indeed, we agree that our data raises a potentially important issue around informed consent. We were only able to make speculations around this issue in our paper, given the nature of the data we had to work with. However, we agree that this is a potentially important consideration that warrants further research and have now added a sentence under 'Implications for research' in the discussion.

Reviewer: 4

Reviewer Name: Emma France

Institution and Country: University of Stirling, Scotland, UK

Please state any competing interests or state 'None declared': none declared

Please leave your comments for the authors below

The manuscript covers an important topic for evidence-based health care - trial retention. As a result of a qualitative systematic review, the authors have provided some useful findings and conclusions for improving trial retention. There are however some important weaknesses in the manuscript which should be addressed.

Response: We were pleased to read that the reviewer thinks our manuscript covers an important topic for evidence-based health care and that as a result of our synthesis we have provided some useful findings and conclusions for improving trial retention.

****Introduction & methods****

1. Why was the chosen qualitative synthesis methodology selected as an appropriate methodology compared to other qualitative synthesis methodologies?

Meta-ethnography is suited to the synthesis of conceptual data – i.e. studies which have developed explanatory concepts, rather than those which just describe what people said/did. It looks like the 8 included studies did not achieve a conceptual level of analysis but provided fairly thin descriptive data making it very difficult to conduct a meta-ethnography and to achieve 'third order' or higher level interpretations.

Response: We consider meta-ethnography a useful approach for interpreting findings across multiple qualitative studies, helping to facilitate the development of new interpretive insights which can be extremely helpful for other researchers and policymakers. Noblit and Hare (1988, P.7) state that "all

synthesis, whether quantitative or qualitative, is an interpretive endeavour. When we synthesize, we give meaning to the set of studies under investigation.” Our understanding is that the interpretive synthesis has the potential to provide a higher level of analysis (than that which is present within individual studies) and this is what we were trying to achieve here in the context of exploring factors involved in trial non-retention. Meta-ethnography is increasingly used within health research. Two members of our team have led and collaborated on published qualitative evidence syntheses (including meta-ethnographies) previously.

****Methods****

2. P 5. The authors say: 'The papers were initially organised in chronological order (but as inductive analysis progressed papers were grouped according to emerging themes).' If each paper contributed to a number of themes how could the whole paper be grouped according to a single theme? How were papers grouped by emerging themes?

Response: We thank the reviewer for raising this important point regarding transparency relating to the translation of studies. See S1 Box for more details of our analytical approach. We have modified S1 Box and added more detail for clarity. The papers were initially organised in chronological order. This is a process that can be a logical and helpful starting point to analysis and is one that has been used previously within other meta-ethnographies. As part of this, we used a standard form which summarised the main themes/concepts, information regarding methods, and any other important information relating to the context of the research within each study (some of this data is illustrated in S1 Table). Although we initially organised papers chronologically in this table, we used this table to facilitate a series of further group discussions around emerging issues and as inductive analysis progressed and we grouped and discussed our data according to the 5 key emerging themes. We have now added an additional table (S3 Table) to show how each paper contributed to a number of themes.

3. P 5. The authors say 'we focused on the findings, concepts and themes used by the papers' authors generating a list of key categories.' This is confusing/ambiguous what does the part 'generating a list of key categories' mean?

Response: Data was extracted from all 11 papers using a standard form which summarised the main themes/concepts, information regarding methods, and any other important information relating to the context of the research (some of this data is illustrated in S1 Table). During this stage, we focussed on both 1st order constructs within included papers (meaning study participant quotations found in the results section of papers) along with 2nd order constructs (meaning the interpretations made by the papers' authors, usually found in the discussion and conclusion sections of papers but also sometimes within the results). Using the standard form, the papers were initially organised in chronological order (but as inductive analysis progressed papers were grouped according to emerging themes) and we focused on the findings, concepts and themes used by the papers' authors generating a list of key categories or themes of our own. This document (along with our other written notes and observations) facilitated discussions at a series of subsequent team meetings and were very useful for consideration of how identified themes from one paper might relate to the others. We have added more detail to S1 Box for clarity.

Study Quality

4. P 5. What is a 'brief' quality assessment? How did you do this and how was CASP used for this e.g. only specific questions used? Why did you decide to do a quality appraisal given they are controversial? The authors have not given details of the results of the quality appraisal.

Response: We accept that the word 'brief' was perhaps misleading and have omitted this as all questions in CASP were used to judge quality. Applying quality criteria to qualitative research remains a contentious issue and there is no consensus regarding whether and how this should be done (Mays 2000; McEwan 2004). However, one author undertook a quality assessment of each of the 11 papers that were identified as being eligible for inclusion in the synthesis (using the 10 item CASP scale). Whilst authors of some qualitative evidence syntheses have chosen to exclude what they deem to be poor quality papers, we made the decision not to exclude any of the identified papers. Although all papers had study aims that were amenable to investigation via qualitative means and all included qualitative data, as a team we deemed some as being richer than others in terms of data and insights (i.e. first and second order constructs). Despite this variation in the overall level of quality, due to the small number of identified studies we considered it more important to retain any relevant findings than disregard based on study quality. In doing so, we would argue that all 11 papers contributed useful elements to the collective whole and enabled us to develop our line of argument in terms of the issues of importance regarding trial non-retention. Please see S1 Box and ENTREQ checklist. For transparency, we have also now added the results of the quality assessment as a supplementary file.

5. There are some ambiguities surrounding the methods in S1 box and it deviates from the meta-ethnography methodology. On the basis of the information provided, what you have done resembles a thematic synthesis or thematic analysis rather than a meta-ethnography.

Response: In S1 Box, the label headings for each of the 7 meta-ethnography steps had been changed in an attempt to make the language more accessible to readers of BMJ Open. However, we accept that in doing so, we perhaps gave the impression that we did not follow the key steps of meta-ethnography. We have amended the text in S1 box accordingly such that it now mirrors the labels of the 7 steps described by Noblit and Hare (1988). The methods and stages employed at each step were guided by the Noblit and Hare text on meta-ethnography and S1 Box highlights these stages. We agree that in a meta-ethnography researchers are indeed aiming to achieve a new interpretation of conceptual data from primary research study accounts/papers. We believe we attempted to achieve this new interpretation from the included primary studies. We have provided additional details of our meta-ethnographic approach in S1 Box for transparency. Hopefully this will clarify that our review of studies was indeed interpretive rather than merely aggregative.

6. S1- the issues around QA being contentious/ having a lack of consensus should be referenced.

Response: Two references have now been added to S1 Box.

7. In S1 Box, the headings used are similar to but different from Noblit and Hare's 7 phases of meta-ethnography. It would be better to be consistent and use all of Noblit and Hare's labels unless there is an important reason for not doing so, Phase 1 = getting started, Phase 2 = deciding what is relevant to the initial interest, Phase 4 should be 'determining how the studies are related' Phase 5 is 'Translating the studies into one another' is and Phase 6 is 'Synthesizing translations'. These phases do overlap and can run in parallel but there are some distinct characteristics.

Response: Headings have now been amended to accurately reflect Noblit and Hare's 7 phases (see S1 Box).

8. Reading the studies - Did all authors read FULL papers or just sections of papers?

Response: All authors read all of the initial 8 included papers. Unfortunately, one of the original authors is currently on sick leave and so two authors (ZS and KG) read the 3 additional papers in full.

9. Phase 4 has not been done in line with the meta-ethnography methodology – it is about more than just ‘identifying’ themes – it involves exploring the relationships between the studies, how they are similar and different. This can be done at the level of concepts/themes from the studies, the underpinning theories, the topic focus, the participants’ characteristics, and so on.

Response: We agree that it involves exploring relationships between studies etc and we did do this as part of our analysis – please see edits to S1 Box.

10. P28. ‘Translating the findings of each study into those of the others’

Authors have not told us what the 14 sub-themes were or how these became 5 broad key themes. Grouping themes – how were they grouped, on what basis? What aspects of the themes from each of the individual studies did you compare and contrast?

Response: Our initial grouping of first and second-order constructs across the papers resulted in 14 sub-themes. From reading and analysing findings from the papers, these were the issues/ideas that we each considered important in terms of things that might make people withdraw from trials. During the process of translating findings from each of the individual studies into those of the others (i.e. comparing and contrasting across studies), following further team discussion these were then grouped and categorised into 5 broad key themes which we interpreted as characterising the main considerations and features that appeared to influence non-retention in the trials under investigation (See S2 Table). Regarding the 14 initial sub-themes - during the initial familiarisation process and team discussions we produced a range of issues that we each considered important in terms of things that might influence whether people withdraw from trials – these formed the original 14 subthemes. Through team reflection and discussion, it was apparent that some of these issues were related or overlapped (e.g. some of the original sub-themes were all broadly about the ‘fit’ of aspects of the trial (or intervention) with individual care and support) and so were reduced, through iterative discussion, to our 5 key themes but whilst still maintaining the integrity of the context of the original study. For transparency, we now provide an illustrative table of the 14 initial sub-themes and illustrate which of the included studies contributed to each sub-theme/main theme (see S3 Table).

11. P28. ‘Synthesising the findings ‘

The description here does not resemble phase 6 of the meta-ethnography methodology (synthesising translations). In this phase you are looking to see if there are common types of translations from phase 5 and to see if some translations or concepts from studies can encompass others. Under phase 6 you could also explain how the ‘line of argument’ was arrived at.

Response: We think that it can be challenging to present the full complexities involved in undertaking qualitative syntheses to the reader in a clear and succinct way. However, we have added more detail for clarity to S1 Box, to illustrate how we approached phase 6.

****Results ****

12. It is not clear what is a novel finding and what's taken directly from the papers so it is not clear if the authors have come up with anything new compared to the findings of the synthesised papers. There needs to be more transparency in what was in the papers and how they each contributed to the findings. Which papers supported and contributed to the 'themes'? For example, you could give a table showing the themes, findings or concepts that were found in each paper.

Response: For transparency, we now provide an illustrative table of the 14 initial sub-themes and demonstrate which of the included studies contributed to each sub-theme/main theme. We also provide illustrative detail re. how papers supported and contributed to our 5 key themes in S2 Table – the extracts in the 2nd column are taken directly from a range of the included studies (appearing as verbatim quotes/text from study authors). Our conceptual interpretations appear in the 1st column.

13. P 6. The authors have not described - What were the 14 subthemes? How (using which criteria/process) were they then grouped into 5 key themes?

This synthesis appears to involve the re-grouping and re-categorising of themes/concepts from the synthesised papers rather than a new interpretation: 'Our initial grouping of first and second-order constructs across the 8 papers resulted in 14 sub-themes. [...]these were then grouped and categorised into 5 broad key themes' (PAGE 28).

Response: Thank you for raising this point of clarification. This is linked to point 10 above – please see previous response.

The themes resulting from the synthesis are descriptive and appear largely based on first order constructs (participants' views and experiences) rather than on second order constructs (the interpretations of the authors of the primary studies).

14. I cannot see any evidence that phase 6 of the meta-ethnography methodology has been conducted or a convincing line of argument synthesis produced (e.g. page 28). It resembles an aggregative review rather than an interpretive review. Meta-ethnography is an interpretive synthesis methodology. There are a number of published guides to selecting a qualitative synthesis methodology e.g. Gough, Thomas, & Oliver. (2012); Noyes & Lewin In Noyes et al (editors). Cochrane handbook, 2011; Paterson (2011) in Hannes & Lockwood; Booth et al 2018 RETREAT guidance.

Response: We agree that in a meta-ethnography researchers are indeed aiming to achieve a new interpretation of conceptual data from primary research study accounts/papers. We believe we attempted to achieve this new interpretation from the included primary studies. We have provided more details of our meta-ethnographic approach in S1 Box. Hopefully this will clarify that our review of studies was indeed interpretive rather than merely aggregative.

15. S2 table is referred to but is missing.

Response: S2 table has been uploaded as a supplementary file.

16. Description of Studies

'Findings were presented from trial non-retainers both before outcome data had been collected (e.g. those who withdrew from the intervention) and/or during the follow up when outcome data was being collected' This is unclear and would benefit from re-wording- findings from where? Do you mean that the papers contained both of these kinds of data, either/or?

Response: We agree this statement is not clear. The papers contained data from people who had dropped out before and/or during follow up – in other words, they included a mix in terms of drop out behaviour. We have amended the manuscript for clarity.

17. The studies were very diverse trials (ankle support/ psycho-education program/aspirin etc) - were retention reasons different?

Response: We accept that the studies are somewhat diverse (although we acknowledge in the manuscript that several have a mental health context and that this might have implications for transferability of findings). However, we found through our line of argument synthesis that themes relating to non-retention were broadly similar across the studies.

18. Figure - how was this figure arrived at? It does not seem to reflect the themes that were developed e.g. sub-optimal informed consent is not a theme but is prominent in the figure. It probably should be a main theme as it's an important finding for trial retention and for ethical trial conduct.

Response: S3 Figure is our attempt at conceptually illustrating the line of argument developed from our synthesis. We agree that our data raises a potentially important issue around informed consent. We made observations in our paper about people perhaps not being as informed as they could be at the consenting stage from our interpretations of withdrawers' quotes (presented in the main manuscript and S2 Table) which illustrate (or at least suggest) that participants discovered things about the intervention post trial entry which made them inclined to withdraw. These observations were implicit as we only had access to the available published data from the 11 studies. These studies explored reasons for drop out as opposed to what went on during recruitment consultations and how this may or may not have impacted on retention. As such, we did not consider this to be a recurring theme within the data but rather a potentially important consideration that warrants further research (please see related comments from Reviewer 2 above).

19. The authors write: 'Our initial grouping of first and second-order constructs across the 8 papers resulted in 14 subthemes. During the process of translating themes from each of the individual studies into those of the others (i.e. comparing and contrasting across studies) these sub-themes were then grouped and categorised into 5 broad key themes which characterised the main considerations and features that appeared to influence non-retention in the trials under investigation.'

19a. How (using which process/criteria) were 1st & 2nd order constructs grouped?

19b. What were the 14 subthemes? 19c. How (using which criteria/ process) were they then grouped into 5 key themes?

Response: Please see our earlier responses to point 10.

20. Were there any refutational/contradictory findings, especially considering the heterogeneity of the included studies e.g. possible cultural differences and different focuses of the trials?

Response: The reviewer raises an important point. We did not find refutational/contradictory findings although we accept that some issues might be more pertinent in certain contexts/ settings (e.g. trials in mental health contexts; drug trials; trials that involve proxy consent etc). We have now added more detail to the 'Strengths and Limitations' section of the discussion. Given the lack of data available to us on possible cultural differences, this was not something that we felt we could highlight in the findings of our manuscript but we mention this as a potential limitation of the synthesis due to the data available in the discussion.

21. page 7. What the authors refer to as line of argument (LOA) appears to be 5 separate descriptive themes not an overarching theory or explanation. For example, how the LOA explains non-retention is under-developed. Some of the themes (experiences of side effects, care preferences, capabilities) are neither about pre-existing beliefs nor life circumstances so are not adequately covered in the LOA.

Response: We have edited our LOA in attempt to make our overarching theory or explanation clearer regarding the factors that influence trial non-retention. Please see page. 6.

(QA) using CASP. Even if you don't exclude studies on the basis of methodological concerns, if you do a QA, you should use this in some way e.g. to inform your discussion, using it to explain variation in findings e.g. describe which papers added most/more/little to the themes.

Response: This has now been added as a supplementary file and we have added more detail around our judgements about quality to the discussion section.

****Discussion****

22. Page 12. The authors need to consider the review and synthesis process in the strengths and limitations and also the limitations of the study accounts/papers synthesised- e.g. missing data on participant characteristics, lack of conceptual findings. International studies were included - is this a strength or a limitation?

Response: We state in the discussion that "Although all papers had study aims that were amenable to investigation via qualitative synthesis and all included qualitative data, some were deemed richer than others in terms of data and insights (i.e. first and second order constructs)." We also refer to the fact that all included studies were from high income countries and indicate that this may have issues re. transferability.

23. Page 13 – the authors say 'we were reassured that the key themes we identified had resonance across all of the 8 included papers to a greater or lesser extent and so are likely to be important considerations within a range of clinical trial contexts.' However, this is not reflected in the results– for the theme 'The compatibility of aspects of trial processes with individual capabilities' – only 3/8 studies contributed and for the theme 'Concerns about or experiences of the trial medication' – only 2/8 studies were relevant and so contributed.

Response: We accept that the wording here was slightly misleading and have edited to read "Although we were reassured that the key themes we identified had resonance across the included papers to a greater or lesser extent and so are likely to be important considerations within a range of clinical trial contexts, some influences on trial non-retention are likely to be more trial specific than others (e.g. concerns about trial medication)."

24. Discussion, page 13 - the authors refer to quality appraisal but don't tell us in the results which papers were rich and how they scored on CASP- this could be added as an additional online file. The authors say 'some were deemed richer than others in terms of data and insights (i.e. first and second order constructs).' How was richness judged? Which papers were rich? Did the richer papers contribute more to the themes?

Response: This has now been added as a supplementary file and we have added more detail around our judgements about quality to the discussion section.

25. Page 13- It looks like the rate of publication of relevant articles is increasing rapidly since 4 new relevant publications were identified as being published between Aug 2016 and October 2017:

'Our original systematic literature search was undertaken in August 2016. Since undertaking and expressing the synthesis within this paper we have updated our search (to October 2017) and have identified 4 additional potentially relevant papers [33,34,35,36].'

This indicates that there are likely to be even more relevant articles now 8 months on that could usefully add to the synthesis which originally identified only 8 articles and which might fill gaps in the data/analysis.

26. There is recent guidance on why & how to update a qualitative synthesis/ meta-ethnography. The author's statement is incorrect: 'there is currently no guidance on how to do this for meta-ethnographies.' (page 13) Please see:

France, E. F., Wells, M., Lang, H., & Williams, B. (2016). Why, when and how to update a meta-ethnography qualitative synthesis. *Systematic reviews*, 5(1), 44.

Response: AS per previous responses we have updated the synthesis. Our search was updated to end of July 2018 and as a result 3 additional papers were identified and have been integrated into the synthesis. The update was informed by the France et al 2016 guidance on updating meta-ethnography and we adopted an 'extend and renovate' model based on their guidance.

The findings of this synthesis are potentially useful. If the authors can revise their description of their methodology to more accurately reflect the analysis they present then I think this paper can make a useful contribution to the trial methods literature. The recent rapid increase in relevant publications would merit an update to include the more recent papers in the synthesis.

VERSION 2 – REVIEW

REVIEWER	Jutta Bleidorn Institute for General Practice, Medical School Hannover, Hannover, Germany
REVIEW RETURNED	25-Sep-2018

GENERAL COMMENTS	Thank you for re-working the paper, it improved much by the expanded literature search until 2018. As for implications for practice and research, please try to focus this part again, to allow the reader to pick up important implications effectively. The first paragraph of practice implications is very good now. The following four paragraphs are still too detailed for an "implication section". Please try to shorten and to reduce to the most important information here.
---

REVIEWER	Tim Rapley Northumbria University, UK
REVIEW RETURNED	02-Jan-2019

GENERAL COMMENTS	I think the tension you face in undertaking this work – and answering some of the questions that seek more conceptual development raised by R4 and myself (R3) – are in part tied to the potential thinness of the papers you are working with. What is clear is that many of the questions we raised in terms of the interpretation and discussion of themes/LOA are not available with this data set. As such, this feels like you need to be very clear
--

	about what is actually missing from the current published work and offer a much stronger directive for the focus of future research.
--	--

REVIEWER	Emma France University of Stirling
REVIEW RETURNED	26-Sep-2018

GENERAL COMMENTS	I appreciate the work that the authors have put in to respond to the reviewers' comments and to update their searches. However, the authors did not always appear to understand the points I was making e.g. including strengths/limitations of the actual process of conducting the synthesis and not just the strengths/limitations of the included primary studies; listing the actual themes/ concepts from the included studies; and clearly stating what's new in your synthesis in addition to what was presented in the included studies. Now that further detail of the methods and included papers have been included, some shortcomings/ gaps of the analysis and synthesis have become clearer. It's still not clear what's new in your synthesis compared to the primary studies, e.g. many of your findings appear in Johansson et al 2015 and Nicholas et al 2010 (two of the richest ones according to your CASP results) but you do not explicitly acknowledge the contribution of individual papers and their concepts/themes to your findings. The updated analytic synthesis appears to have been somewhat selective about which findings it has included in the synthesis. *Abstract* Error in abstract- 'The studies were undertaken between 2008 and 2015.' should be 2008-2018 *Methods* Page 4, line 3- there seems to be a gap in the search dates- is this an error? Were no searches done between Aug 2016 and Oct 2017? Postel et al 2010 doesn't appear to be a qualitative study at all but a structured questionnaire of drop out reasons- why was it included? You even say it was 'not a qualitative study in your table of CASP appraisal results. *Results* The analytic synthesis is not quite there yet in terms of sophistication and nuance and there are some doubts about its comprehensiveness in terms of representing the included studies. You haven't actually shown all the themes/ concepts from each of the included papers - a list of these would have been more useful than table S2 which just repeats what's already in the findings section rather than adding anything new. Theme 1A –an illustrative quote refers to the participant who just didn't want to be reminded of alcoholism. This is more than just being 'too well' to engage – it's about not wanting to be reminded of addiction when you are moving on and recovering . This resonates with the theme in Nicholas et al 2019 of 'did not want to think about illness (confronting/overwhelming)' which doesn't really seem to appear in your analysis. Your analysis hasn't really paid attention to the relationships between themes within papers e.g. Johansson et al 2015 found an interaction between patient factors and treatment factors . Themes
--

	from a paper shouldn't just be viewed in isolation from one another. Table S3 now shows the 14 initial sub-themes and which papers contributed data to these- this is a helpful addition. From this table the strongest themes (i.e. having support from the largest number of studies) appear to be those that are unlikely to be modifiable or very difficult to modify as they relate to the key theme 1 (current health state) & key theme 5 (broader lives/lack of fit): perceptions of being too well (supported by 7 papers), aspects of life getting in the way (supported by 8 papers) – this has implications for some of your conclusions which focus on modifiable aspects. Some themes from the 3 included papers identified in the updated searches are not accounted for in your updated analysis - this raises questions about its thoroughness and comprehensiveness. Sari 2017 (paper in your update)- found social barriers such as dropping out due to a lack of supportive family members and it's also not clear where Sari's themes of fear, guilt and shame come into your updated analysis – this seems new compared to what you found in the original synthesis. Henshall 2018 report a reason for drop out as being allocated to the control arm when you'd rather have been in the intervention arm - this is not represented in the updated synthesis and has implications for the conclusions e.g. they shouldn't have been recruited to trial if they had a strong preference for getting the intervention rather than the control. Theme 2. The 'fit of aspects of the trial with individual preferences for care and support) The theme title is misleading- this theme is about aspects of the INTERVENTIONS being trialed e.g. content of modules, not trial processes and procedures e.g. randomisation or consent processes. Aspects of non-medication treatments, e.g. therapy, exercise, are not well represented –and could have usefully been combined with medication problems in theme 4 e.g. as treatment problems *Discussion* It is often really hard to access and interview 'drop outs' due to both ethical barriers but also to difficulties recruiting people who've disengaged from a study. This could help explain why few studies were identified and could make further primary research in this area challenging. It would be worth acknowledging this point. You often conflate 'trial' with 'intervention' – the trial is the study design and processes, the intervention is what's delivered and tested. E.g. 'The compatibility or otherwise of various trial processes with individual capabilities...' (p64, line16) - I think capabilities were raised more in relation to the nature of the interventions, e.g. content too complex, too cognitively demanding. Also p12- refers to 'trial burden' but you are actually referring to intervention burden . You need to be careful to distinguish between trial processes/ trial design and intervention design/aspects. It appears that many of the issues relate NOT to the trial itself but to the interventions- this raises important lessons for INTERVENTION design and delivery rather for TRIAL design and delivery.
--	--

VERSION 2 – AUTHOR RESPONSE

Reviewers' comments	Response
Reviewer: 2 Reviewer Name: Jutta Bleidorn Thank you for re-working the paper, it improved much by the expanded literature search until 2018. As for implications for practice and research, please try to focus this part again, to allow the reader to pick up important implications effectively. The first paragraph of practice implications is very good now. The following four paragraphs are still too detailed for an "implication section". Please try to shorten and to reduct to the most important Information here.	We thank the reviewer for their comments and we are pleased to read that they believe our paper is much improved by the expanded literature search. We have now re-structured and edited the implications for practice and research sections of the discussion in an attempt to allow the reader to pick up important implications effectively.
Reviewer: 4 Reviewer Name: Emma France 1. I appreciate the work that the authors have put in to respond to the reviewers' comments and to update their searches. However, the authors did not always appear to understand the points I was making e.g. including strengths/limitations of the actual process of conducting the synthesis and not just the strengths/limitations of the included primary studies; listing the actual themes/ concepts from the included studies; and clearly stating what's new in your synthesis in addition to what was presented in the included studies.	We thank the reviewer for their constructive comments. In addition to outlining the limitations of the included primary studies, we have now added detail to the discussion outlining the strengths/limitations of the actual process of conducting the synthesis. We provide detailed responses to the other points raised (i.e. listing themes/concepts from the included studies; stating what's new in addition to what was presented in the included studies) in the relevant sections below.
2. Now that further detail of the methods and included papers have been included, some shortcomings/ gaps of the analysis and synthesis have become clearer. It's still not clear	We apologise that the reviewer believes it is difficult to identify what's new in our synthesis and what each study contributes. We would argue that we do acknowledge the contribution

what's new in your synthesis compared to the primary studies, e.g. many of your findings appear in Johansson et al 2015 and Nicholas et al 2010 (two of the richest ones according to your CASP results) but you do not explicitly acknowledge the contribution of individual papers and their concepts/themes to your findings. The updated analytic synthesis appears to have been somewhat selective about which findings it has included in the synthesis.	of individual papers and their concepts/themes contributions to our findings. We do this throughout the results section of the paper (by referencing every time one of our observations appear in a paper) and also within Tables S2 and S3. Many of our findings appear in Johansson et al 2015 and Nicholas et al 2010 (see S3 Table), but not all (e.g. for Nicholas, not all of Theme 2 and none of Themes 3 or 4; for Johansson , not all of Theme 2 and none of Theme 4). Given that they are two of the richest included studies it is perhaps not surprising that they contribute more – but they do not contribute to all our insights. Indeed, none of our included studies contained every single theme/sub-theme we identified. What we have attempted to do in this synthesis is specifically look for patterns and connections across the whole dataset to allow us to more fully represent and understand what the key factors impacting on retention in trials might be. Regarding the last point here about our update – please see details below in the relevant section.
3. *Abstract* Error in abstract- 'The studies were undertaken between 2008 and 2015.' should be 2008-2018	Thank you – we have amended this.
4. *Methods* Page 4, line 3- there seems to be a gap in the search dates- is this an error? Were no searches done between Aug 2016 and Oct 2017?	This was an error. It has now been amended to August 2016- July 2018 (updated search).
5. Postel et al 2010 doesn't appear to be a qualitative study at all but a structured questionnaire of drop out reasons- why was it included? You even say it was 'not a qualitative study in your table of CASP appraisal results.	We agree with the reviewer that Postel et al 2010 is not a well-designed qualitative study in that the only qualitative data is from open-ended questions in an online questionnaire. However, responses to general open questions within questionnaires can be considered to be

	qualitative data (Stecker 1992). We have edited our paper in places throughout to reflect this. We chose to include Postel et al 2010 because given the number of eligible studies in this synthesis was so small, we felt it would be more helpful to include the qualitative data within Postel et al 2010 (and recognise limitations) rather than exclude. We state this in the strengths and limitations section of the discussion.
6. You haven't actually shown all the themes/ concepts from each of the included papers - a list of these would have been more useful than table S2 which just repeats what's already in the findings section rather than adding anything new.	The reviewer is correct that we haven't included a list of all of the original themes/concepts from each of the included studies. We believe it would be unusual to list all of the original themes/concepts from the included studies. S3 Table provides the themes/concepts we generated in our team discussions and how they fit across the studies – providing further transparency about where our interpretations and overall model came from. S2 table provides new raw data from the included studies that is not in the results. In addition to S2 and S3 Tables we have also provided key contextual information about each of the included papers in S1 Table – this is in line with recommended reporting criteria in recently published guidance re. reporting of meta-ethnography published by the Reviewer (France et al 2018).
7. Theme 1A –an illustrative quote refers to the participant who just didn't want to be reminded of alcoholism. This is more than just being 'too well' to engage – it's about not wanting to be reminded of addiction when you are moving on and recovering. This resonates with the theme in Nicholas et al 2019 of 'did not want to think about illness (confronting/overwhelming)' which doesn't really seem to appear in your analysis.	We agree that , the first part of this quote is about not wanting to be reminded, but the second part relates to the participant saying that she does not want to be reminded because she no longer thinks she has an alcohol problem (i.e. she feels that she has suitably recovered – 'too well' to engage). However, we do agree that it does resonate with the theme in Nicholas et al 2019, and so we have incorporated a sentence to clarify (see results section Theme 1A).
8. Your analysis hasn't really paid attention to the relationships between themes within papers e.g. Johansson et al 2015 found an interaction between patient factors and treatment factors. Themes from a paper shouldn't just be viewed in isolation from one another.	We are disappointed that the reviewer believes we have not paid attention to relationships between themes within papers. We would argue that we did pay due attention to the relationships between findings within papers and certainly this featured heavily in our discussions during inception of our synthesis. We make the point in

	our paper (in the results and again in the discussion) that the issues we identified were not necessarily mutually exclusive but were rather a synergistic combination of factors that could apparently work towards trial non-retention – implicitly highlighting the interconnection between the themes within studies. We also make the point in the discussion that there can be a complex inter-relationship between participant characteristics and trial processes/factors. In addition to this, given the reviewers suggestion in point 10 below, we have now included references to emotional influences across several of the themes. This provides further illustration of the relationships between themes identified.
9. Table S3 now shows the 14 initial sub-themes and which papers contributed data to these- this is a helpful addition. From this table the strongest themes (i.e. having support from the largest number of studies) appear to be those that are unlikely to be modifiable or very difficult to modify as they relate to the key theme 1 (current health state) & key theme 5 (broader lives/lack of fit): perceptions of being too well (supported by 7 papers), aspects of life getting in the way (supported by 8 papers) – this has implications for some of your conclusions which focus on modifiable aspects.	We are pleased the reviewer finds our addition helpful. However, we were not trying to suggest that all themes we identified were modifiable - but rather the way in which a trial is presented to individuals needs to take account of these influencing factors. We would argue that trialists need to think carefully about how the design of their trial might contribute to non-retention and that there is potential to modify trial design to improve retention – trialists need to think about the demands of their trial in terms of what they are asking participants to do – e.g. multiple follow up questionnaires; complicated workbooks for people who are mentally unwell, demanding exercise regimes etc. All of our identified themes talk to the need for improved communication during informed consent consultations and ongoing communication with participants over time – i.e. by making trial expectations clearer to enable participants to begin to address how it might fit into their lives (or not). We know from anecdotal evidence from our CTU that one of the reasons for participants not returning questionnaires relates to them feeling

	better. We also know that the importance of retention for clinical trials is not usually explained to participants at the time of recruitment (we have a paper currently under preparation that provides evidence that issues around retention are largely undiscussed during trial consultations). We have now included a few sentences in the discussion section to highlight that whilst not all the factors we identified are modifiable there influence needs to be recognised.
10. Some themes from the 3 included papers identified in the updated searches are not accounted for in your updated analysis - this raises questions about its thoroughness and comprehensiveness. Sari 2017 (paper in your update)- found social barriers such as dropping out due to a lack of supportive family members and it's also not clear where Sari's themes of fear, guilt and shame come into your updated analysis – this seems new compared to what you found in the original synthesis. Henshall 2018 report a reason for drop out as being allocated to the control arm when you'd rather have been in the intervention arm - this is not represented in the updated synthesis and has implications for the conclusions e.g. they shouldn't have been recruited to trial if they had a strong preference for getting the intervention rather than the control.	We thank the review for highlighting this discrepancy. With regard to social barriers, we had added non-retention due to lack of support from family members from Sari 2017 to Theme 5, as we felt it related to aspects to their life that made it hard for them to engage. However, on reflection, although we still think it broadly fits within this theme, it relates specifically to existing social networks and so we have added some additional clarification to this section in the results. All other included studies were re-checked (by 2 authors re-reading full text) to check for any other relevant data relating to social barriers. All other relevant data had already been included. The second point raised by the reviewers with regard to Sari's mention of 'fear, guilt and shame' within their paper related to participants feeling that they were too overweight and unfit to participate in a group exercise intervention has also been added to the synthesis. However, we believe this 'fear, guilt' and shame' aspect is part of a broader sub-theme related to emotional influences that is woven across several of the other themes (which became apparent on re-reading all included studies with this reviewer-led focus on emotional influence). We have now incorporated this aspect where relevant. Our re-working of the synthesis to accommodate this reviewer request also highlights one of the limitations of qualitative evidence synthesis which we now also point to

	in the limitations section, that different review teams may interpret data in these types of evidence synthesis in slightly different ways due to pre-existing world views or expertise across research areas. With regard to the . Henshall 2018 paper. This paper included data from both trial completers and people who had dropped out and the data the reviewer refers to about participants wanting to be in the intervention arm was from two people who completed the trial (therefore we excluded this data from our synthesis).
11. Theme 2. The 'fit of aspects of the trial with individual preferences for care and support) The theme title is misleading- this theme is about aspects of the INTERVENTIONS being trialled e.g. content of modules, not trial processes and procedures e.g. randomisation or consent processes.	We thank the reviewer for highlighting this important point in relation to a difference between trial intervention and trial process. However, we disagree with this point. We believe it is important in our synthesis to keep the theme label as it is to ensure the interpretation can relate to either intervention or process as this is reflective of the data from the included studies. From some of the quotes (e.g. "I would have liked to have more of a personal contact, it became a little distant everything, to do on the internet, because it is so heavy stuff, it's nice to meet a real person when you're working with heavy things like this" [24, no gender or age details, generalised anxiety disorder]'), it is not clear if the interviewee is talking only about the intervention or the follow up or both (as most CBT trials are delivered and followed up remotely) so we would be making assumptions about whether they are solely referring to the intervention or to follow up processes. Also, many trial participants (as we are seeing evidenced in this synthesis) are unaware that the trial is more than the intervention i.e. they conflate trial process with intervention – so we cannot assume that the nuances of differentiating intervention from follow-up are appreciated. As such, we would prefer to leave as "fit of aspects of the trial with individual preferences for care and support"

12. Aspects of non-medication treatments, e.g. therapy, exercise, are not well represented –and could have usefully been combined with medication problems in theme 4 e.g. as treatment problems	We appreciate the reviewers comment however, we would argue that aspects relating to views about the therapy, exercise etc are well represented throughout the results, especially when we reflect on the various preferences people have for these non-medication treatments. We accept that there is some overlap between the themes we identified but feel strongly that this theme is distinguishable in the side effects relating explicitly to the trial intervention (medication) and so important to highlight. This is again linked to the previous comment on conflating process with intervention – this is one of the few that doesn't and therefore important to keep separate.
13. It is often really hard to access and interview 'drop outs' due to both ethical barriers but also to difficulties recruiting people who've disengaged from a study. This could help explain why few studies were identified and could make further primary research in this area challenging. It would be worth acknowledging this point.	We agree and had acknowledged this in our previous manuscript version within 'Implications for research.' We have also now added a sentence to the 'strengths and limitations' section of the discussion to reflect this.
14. You often conflate 'trial' with 'intervention' – the trial is the study design and processes, the intervention is what's delivered and tested. E.g. 'The compatibility or otherwise of various trial processes with individual capabilities...' (p64, line16) - I think capabilities were raised more in relation to the nature of the interventions, e.g. content too complex, too cognitively demanding.	For the same reasons provided earlier, we would like to leave our wording as it is – i.e. it is not always clear whether participants are discussing the intervention or aspects of the follow up.
15. Also p12- refers to 'trial burden' but you are actually referring to intervention burden. You need to be careful to distinguish between trial processes/ trial design and intervention design/aspects. It appears that many of the issues relate NOT to the trial itself but to the interventions- this raises important lessons for INTERVENTION design and delivery rather for TRIAL design and delivery.	Please see above response.
Reviewer Name: Tim Rapley	We thank the review for this comment and agree with the points raised. We have added mention of this to the 'Strengths and Limitations' section of the discussion. We have also edited

I think the tension you face in undertaking this work – and answering some of the questions that seek more conceptual development raised by R4 and my myself (R3) – are in part tied to the potential thinness of the papers you are working with. What is clear is that many of the questions we raised in terms of the interpretation and discussion of themes/LOA are not available with this data set. As such, this feels like you need to be very clear about what is actually missing from the current published work and offer a much stronger directive for the focus of future research.	the 'Implications for research' section in an attempt to address these points and to provide a stronger directive for future research.
---	---

VERSION 3 – REVIEW

REVIEWER	Tim Rapley Northumbria University, UK
REVIEW RETURNED	11-Mar-2019

GENERAL COMMENTS	I feel you have responded to our comments - and as reviewers, we always feel more could be done to refine, adjust - but I think this fine, given the data set you are working with. Just some minor editing/spelling corrections Page 10: with participants in one of these studies staing that they felt unintelligent because of their inability to understand [24]. Shoud be 'stating' Page 14: This finding is supported by arecent study of patient information documentation from UK NIHR funded trials that has highlighted that withdrawal and retention are poorly described and that statements about the value of retention are infrequent [39] 'A recent' Page 19- delete 'Authors: Dan Brunsdon; Linda Biesty; Peter Brocklehurst;
--

VERSION 3 – AUTHOR RESPONSE

We are pleased that the reviewers have recommended publication of our paper. We are pleased to submit a revised version of our manuscript which takes account of the latest set of minor comments

from 1 of the reviewers. We have corrected the typo on p.10 (to stating); we have edited the sentence on p.14 to read 'A recent'. The reference on p.19 has now been published and so we have kept the author listing, but have now added the paper DOI to the reference. We look forward to hearing from you.